# Pulmonary immune cell trafficking promotes host defense against alcohol-associated *Klebsiella* pneumonia

Derrick R. Samuelson [1,2✉], Min Gu[1], Judd E. Shellito[1,3], Patricia E. Molina [4], Christopher M. Taylor [3], Meng Luo[3] & David A. Welsh [1,3]

The intestinal microbiota generates many different metabolites which are critical for the regulation of host signaling pathways. In fact, a wide-range of diseases are associated with increased levels of local or systemic microbe-derived metabolites. In contrast, certain bacterial metabolites, such as tryptophan metabolites, are known to contribute to both local and systemic homeostasis. Chronic alcohol consumption is accompanied by alterations to intestinal microbial communities, and their functional capacities. However, little is known about the role of alcohol-associated dysbiosis on host defense against bacterial pneumonia. Our previous work using fecal transplantation demonstrated that alcohol-associated intestinal dysbiosis, independent of ethanol consumption, increased susceptibility to *Klebsiella* pneumonia. Here, we demonstrate that intestinal microbiota treatments mitigate the increased risk of alcohol-associated pneumonia. Treatment with the microbial metabolite indole or with probiotics reduced pulmonary and extrapulmonary bacterial burden, restored immune responses, and improved cellular trafficking required for host defense. Protective effects were, in part, mediated by aryl hydrocarbon receptors (AhR), as inhibition of AhR diminished the protective effects. Thus, alcohol appears to impair the production/processing of tryptophan catabolites resulting in immune dysregulation and impaired cellular trafficking. These data support microbiota therapeutics as novel strategies to mitigate the increased risk for alcohol-associated bacterial pneumonia.

[1] Department of Internal Medicine, Section of Pulmonary/Critical Care & Allergy/Immunology, Louisiana State University Health Sciences Center, New Orleans, LA, USA. [2] Department of Internal Medicine, Division of Pulmonary, Critical Care, & Sleep, University of Nebraska Medical Center, Omaha, NE, USA. [3] Department of Microbiology, Immunology and Parasitology, Louisiana State University Health Sciences Center, New Orleans, LA, USA. [4] Department of Physiology, Louisiana State University Health Sciences Center, New Orleans, LA, USA. ✉email: derrick.samuelson@unmc.edu

Alcohol use disorder (AUD) and respiratory infections are major global health burdens[1,2]. AUDs are an established risk factor for bacterial pneumonia[3]. Patients with AUDs are more frequently infected with highly virulent respiratory pathogens, and experience increased morbidity and mortality from these infections when they occur. *Klebsiella pneumoniae* infections are overrepresented in pneumonia patients with AUD[3,4], and AUD patients admitted to the hospital with community-acquired *Klebsiella* pneumonia experience almost double the mortality of AUD patients infected with other pathogens[4].

There are several potential mechanisms by which AUD increase the risk of pneumonia. These include aspiration of microbes from the upper alimentary tract, decreased mucus-facilitated clearance of bacterial pathogens from the upper airway, and impaired pulmonary host defenses[5]. In addition, the prevalence of oropharyngeal colonization with *K. pneumoniae* may be as much as four times higher in patients with AUD compared with non-AUD patients[6]. The colonization by pathogenic organisms, combined with the acute intoxicating effects of alcohol and the subsequent depression of the normally protective gag and cough reflexes, leads to more frequent and severe pneumonias due to Gram-negative organisms. Further, chronic alcohol feeding decreases the number of circulating lymphocytes and impairs immune responses, such as Th1 and Th17 responses to various challenges[7–9]. Together, impaired innate and adaptive immunity, and increased pathogen colonization collectively increase the risk of pneumonia in AUD patients[3,10,11].

Chronic alcohol ingestion leads to bacterial overgrowth and dysbiosis in the small and large intestinal tract of animals and humans[12–16]. The increase in both aerobic and anaerobic bacteria is most pronounced in the proximal small intestine in alcohol-fed mice[16], but also extends to the large intestine as early as 1 week after alcohol feeding[15]. Chronic ethanol feeding also reduces; amino acid metabolism, levels of short-chain, and saturated long-chain fatty acids[17–19], and bile acid metabolism in the gut[20]. Together, these data demonstrate that chronic alcohol use is associated with impairments in mechanisms that maintain intestinal microbiota homeostasis. While the taxonomic composition of the alcohol-associated gut microbiome has been characterized, we are just beginning to understand the functional consequences of alcohol-dysbiosis. However, advances in metabolomics have helped further our understanding of the mechanisms by which intestinal dysbiosis affect distal organ systems. Specifically, bacterial products (e.g., LPS, flagellin, bacterial DNA, peptidoglycans) cross the epithelial barrier, enter the systemic circulation, and trigger a potent inflammatory response[21]. Yet, certain bacterial metabolites are known to contribute to both local and systemic homeostasis. For example, indole derivatives of tryptophan catabolism by bacterial species have been shown to play an important role during many chronic diseases[22–26].

Preclinical evidence demonstrates that the intestinal microbiota plays a crucial role in the immune response to bacterial and viral respiratory infections[27–30]. We have previously shown that alcohol-associated dysbiosis, independent of ethanol, increased susceptibility to *Klebsiella* pneumonia[31]. Increased susceptibility was associated with increased intestinal permeability and altered immune cell numbers in both the lungs and intestinal tract. Similarly, we found that microbial products associated with alcohol-induced dysbiosis increased permeability, as well as immune activation in an ex-vivo model system[32]. However, it is not clear how altered microbial products contribute to alcohol-associated pneumonia, or whether restoration of microbial metabolites could mitigate alcohol-associated pneumonia. This study sought to investigated, whether targeting the intestinal microbiota could mitigate the increased risk of alcohol-associated pneumonia via alterations to the composition of the microbiota and cellular trafficking.

## Results

**Microbiota supplementation mitigates alcohol-associated pneumonia**. Our previous data suggest that therapeutic targeting of the intestinal microbiota during alcohol-consumption could mitigate alcohol-associated impairments in host defense. Here, we sought to investigated whether microbial supplementation could mitigate the increased risk of alcohol-associated pneumonia. We chose two microbiota-targeting approaches for the prevention of alcohol-associated pneumonia: (1) a probiotic cocktail (containing *Bifidobacterium bifidum, Bifidobacterium lactis, Lactobacillus plantarum, Lactobacillus reuteri*, and *Lactobacillus rhamnosus*), and (2) the microbial specific tryptophan catabolite indole. Our experimental model is shown in Fig. 1a. Neither alcohol feeding nor treatments influenced body weight (Supplemental Fig. 1a), daily food intake (Supplemental Fig. 1b), or blood alcohol levels (Supplemental Fig. 1c). We first assessed the effects of microbiota-targeting treatments on susceptibility to *K. pneumoniae* in alcohol-fed mice. Alcohol-fed mice had a higher pulmonary burden of *K. pneumoniae*, after 48 h of post-infection, compared to control mice (Fig. 1b), and increased dissemination of pulmonary infection compared to control mice (Fig. 1c). Treatment with indole or probiotics reduced pulmonary and extra-pulmonary *K. pneumoniae* burden in alcohol-fed mice (Fig. 1b, c). As alcohol-feeding is known to cause increased epithelial permeability and decreased numbers of mucosal immune cells[8,23], we examined mucosal permeability as well as pulmonary and intestinal immune responses following *K. pneumoniae* infection. Alcohol-fed mice exhibited marked increases in pulmonary and intestinal permeability as measured by circulating levels of surfactant protein-D (SDP-1; a recognized biomarker of lung damage[33]) and intestinal fatty-acid binding protein (IFABP; a recognized biomarker of intestinal damage[34,35]), respectively (Fig. 2a, b) suggesting an increase in pulmonary and intestinal permeability. However, in alcohol-fed mice indole or probiotic treatment reduced both SDP-1 and IFABP to levels that were indistinguishable from control-fed mice after infection with *K. pneumoniae* (Fig. 2a, b). Finally, we evaluated the number of pulmonary and intestinal immune cell 48 h post-infection. Our flow cytometry gating strategy is shown in Fig. 2c. There was a marked decrease in the number of immune cells in the lungs of alcohol-fed mice. In contrast, we found an increase in the number of intestinal immune cells in alcohol-fed mice compared to control-fed mice (Fig. 2d, e). Similar to the results found for epithelial permeability, both pulmonary and intestinal immune cell numbers were restored to control-fed levels in alcohol-fed mice treated with either indole or the probiotic cocktail (Fig. 2d, e).

**Microbiota supplementation mitigates alcohol-associated pneumonia via aryl hydrocarbon receptor signaling**. The intestinal microbiota modulates aryl hydrocarbon receptors (AhR) signaling via production of tryptophan catabolites[26]. We sought to investigate the role of AhR in indole and probiotic mediated mitigation of alcohol-associated pneumonia, by treatment with the AhR inhibitor CH-223191. Additionally, we utilized heat-killed probiotics to determine if viable probiotics are required for the protective effects or if the microbial components of the probiotics are sufficient. The experimental model/timeline is shown in Fig. 3a. Neither alcohol feeding nor treatments had an effect on body weight (Supplemental Fig. 2a), daily food intake (Supplemental Fig. 2b), or blood alcohol levels (Supplemental

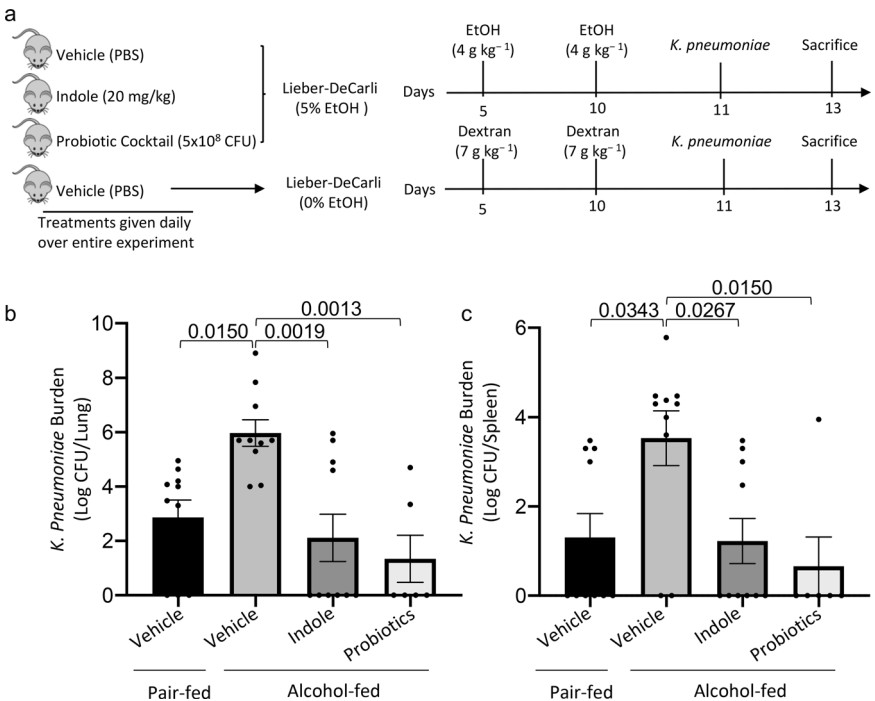

**Fig. 1 Microbiota supplementation mitigates alcohol-associated pneumonia.** Binge-on-chronic alcohol-fed mice with and without treatment were infected with *Klebsiella* and sacrificed 48 h post-infection. **a** Experimental schema. **b** *Klebsiella* lung burden (Log CFU/ml). **c** *Klebsiella* splenic burden (Log CFU/ml). Bars represent the mean ± SEM and dots represent individual mice. *P* values are indicated in the figure and were determined by one-way ANOVA with Sidak's multiple comparison test. $N = 10$/group.

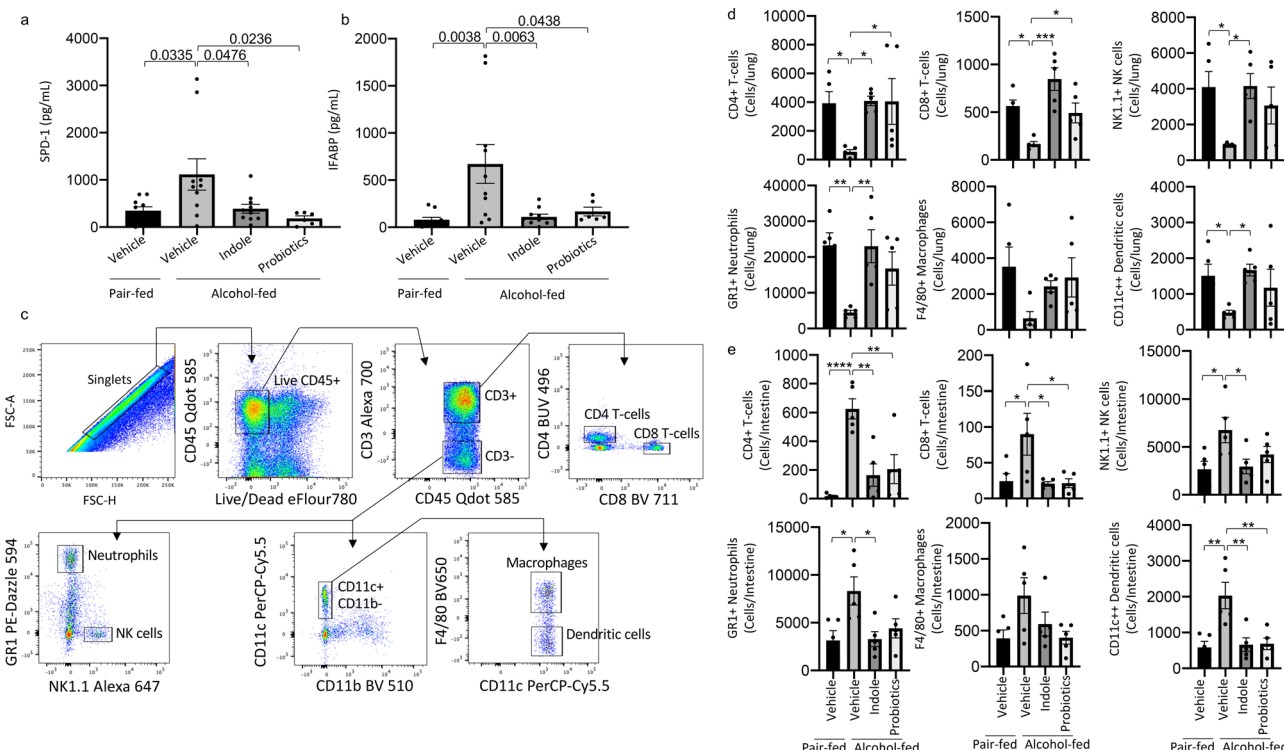

**Fig. 2 Microbiota supplementation mitigates alcohol-associated permeability and immune dysfunction.** Binge-on-chronic alcohol-fed mice with and without treatment were infected with *Klebsiella* and sacrificed 48 h post-infection. **a** Levels of circulating surfactant protein D1 (SPD-1). **b** Levels of circulating intestinal fatty acid binding protein (IFABP). **c** Flow cytometry gating strategy. **d** Number of pulmonary immune cells in binge-on-chronic alcohol-fed mice with and without treatment infected with *Klebsiella*. **e** Number of intestinal immune cells in binge-on-chronic alcohol-fed mice with and without treatment infected with *Klebsiella*. Bars represent the mean ± SEM and dots represent individual mice. *P* values are indicated in the figure or denoted by *$P < 0.05$, **$P < 0.001$, ***$P < 0.0001$, ****$P < 0.00001$, $^\$P < 0.000001$, as determined by one-way ANOVA with Sidak's multiple comparison test. $N = 10$/group.

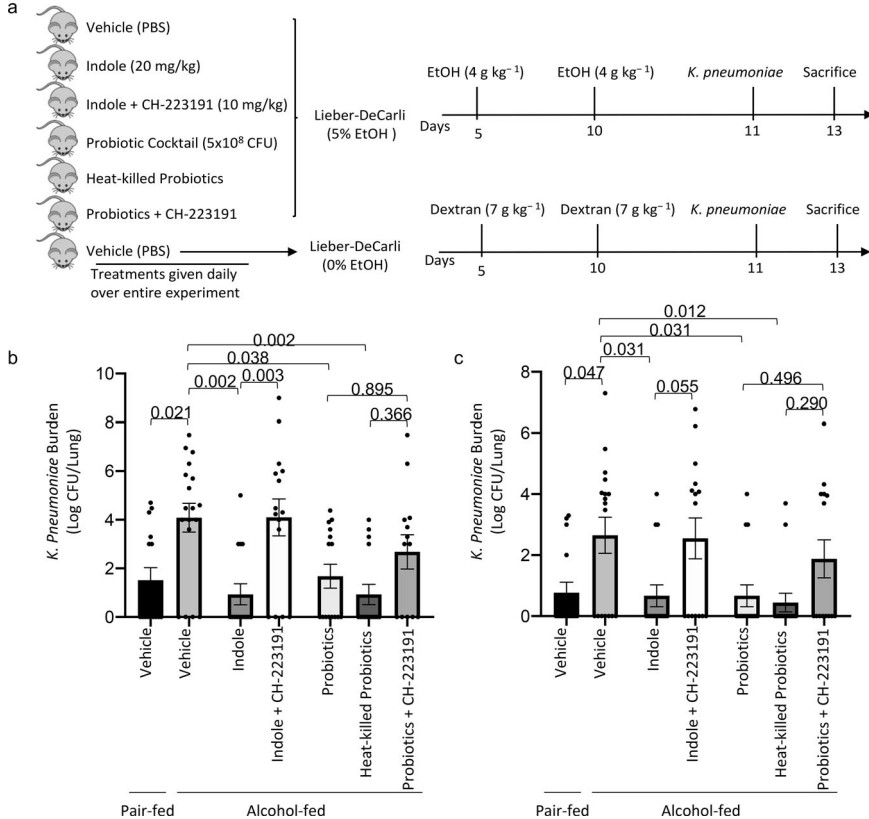

**Fig. 3 Indole and probiotic treatment mitigates alcohol-associated pneumonia via aryl hydrocarbon receptor signaling.** Binge-on-chronic alcohol-fed mice with and without treatment were infected with *Klebsiella* and sacrificed 48 h post-infection. **a** Experimental schema. **b** *Klebsiella* lung burden (Log CFU/ml). **c** *Klebsiella* splenic burden (Log CFU/ml). Bars represent the mean ± SEM and dots represent individual mice. *P* values are indicated in the figure and were determined by one-way ANOVA with Sidak's multiple comparison test. $N = 10$/group.

Fig. 2c). Similar to the previous experiment, alcohol-fed mice had a higher pulmonary burden of *K. pneumoniae*, as well as extra-pulmonary *K. pneumoniae* after 48 h of infection compared to control mice (Fig. 3b, c, respectively). Treatment with indole or probiotics reduced pulmonary and extra-pulmonary *K. pneumoniae* burden in alcohol-fed mice (Fig. 3b, c, respectively), while inhibition of AhR mitigated the protective effects of indole. Probiotic-mediated effects were less affected by AhR inhibition. There was an overall reduction in pulmonary burden that failed to reach statistical significance, suggesting that probiotics may target multiple pathways along the gut-lung axis. Forty-eight hours post *K. pneumoniae* infection, alcohol-fed mice exhibited marked increases in pulmonary and intestinal permeability as measured by circulating levels of SDP-1 and IFABP, respectively (Fig. 4a, b). However, in alcohol-fed mice treated with indole or with the probiotic cocktail pulmonary and intestinal permeability were reduced and were indistinguishable for control fed mice. Similarly, heat-killed probiotics provided protection against alcohol-associated pulmonary and intestinal damage. The protective effects of indole or probiotics was dependent on AhR signaling, as inhibition of AhR ablated the protective effects (Fig. 4a, b). We then evaluated the number of pulmonary (Fig. 4c) and intestinal (Fig. 4d) immune cell 48 h post-infection, and found a decrease in the number of immune cells in the lungs but an increase in the number of intestinal immune cells in alcohol-fed mice, compared to control-fed mice. However, both pulmonary and intestinal immune cell numbers were restored to control-fed levels in alcohol-fed mice treated with either indole or the probiotic cocktail (Fig. 4c, d, respectively). Immune cell numbers were not different in alcohol-fed mice treated with heat-killed probiotics or live probiotics. The protective effects of indole

or probiotics were likewise dependent on AhR signaling, as inhibition of AhR prevented normalization of immune cell numbers (Fig. 4c, d).

Alcohol consumption has been previously shown to decrease IL-22 expression/secretion in the intestinal tract of mice[23,36]. The intestinal microbiota can modulate IL-22 production via production of tryptophan catabolites, which are ligands for aryl hydrocarbon receptors (AhR)[26]. As such, we sought to investigate the expression/secretion of IL-22 in indole and probiotic-treated mice. Similar to the previous experiment, alcohol-fed mice had a higher pulmonary burden of *K. pneumoniae* 48 h post-infection compared to control mice (Fig. 5a). Treatment with indole or probiotics reduced pulmonary *K. pneumoniae* burden in alcohol-fed mice (Fig. 5a), while inhibition of AhR mitigated the protective effects of indole. We also evaluated the frequency of IL-22-expressing pulmonary and intestinal immune cells (gated on CD3− AhR+). Gating strategy for IL-22-expressing cells is shown in Fig. 5b. The percent of IL-22-expressing cells was reduced in ethanol-fed mice compared with controls mice (Fig. 5c, d). Further, treatment with indole or probiotics increased the frequency of IL-22+ cells in the lungs and intestines of alcohol-fed mice (Fig. 5c, d), while inhibition of AhR mitigated the protective effects.

In addition, we further confirmed the role of AhR and IL-22 in our model by evaluating the gene expression of *Ahr*, *Cyp1a1* and *Il22* in the lung and small intestine of alcohol-fed and treated mice. AhR activation regulates the expression of both Cyp1a1 and IL-22[37]. Ethanol-fed mice had a decrease in level of *Ahr*, *cyp1a1*, and *Il22* mRNA in the lungs and intestines compared to pair-fed mice (Fig. 6a–f). Similar to our flow cytometry data, the expression of *Ahr*, *cyp1a1*, and *Il22* was restored to levels similar

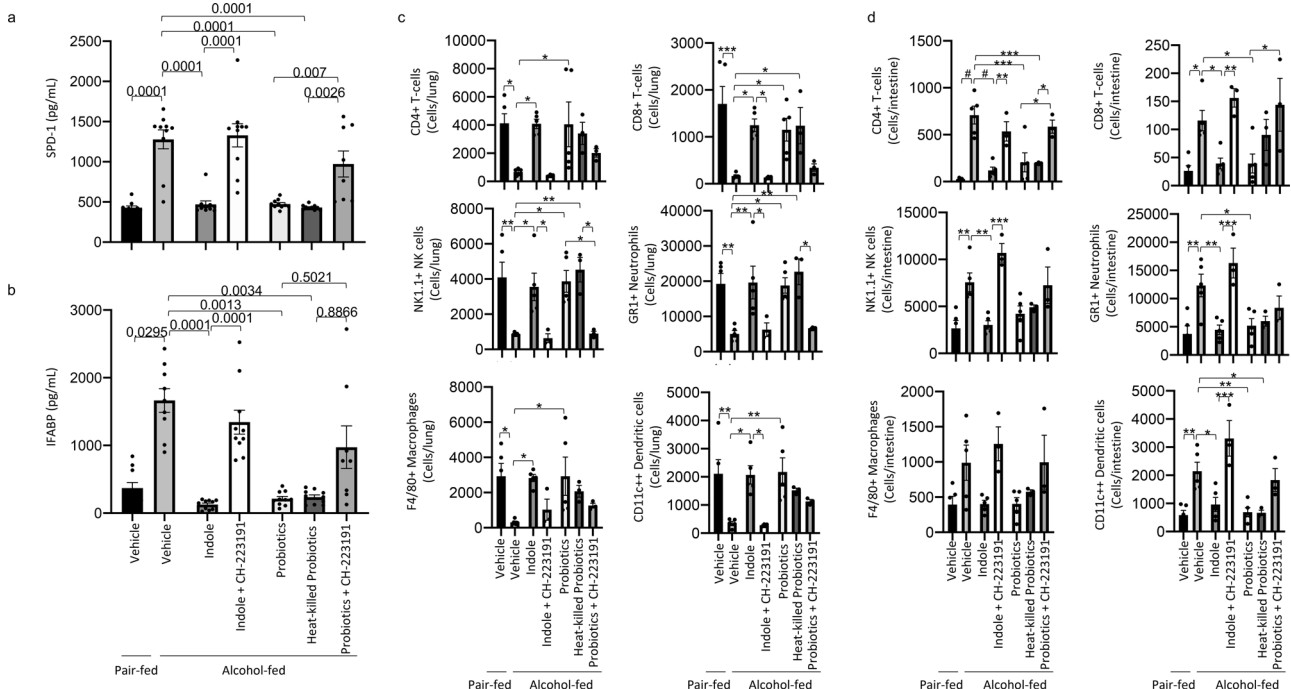

**Fig. 4 Indole and probiotic treatment mitigates alcohol-associated permeability and immune dysfunction via aryl hydrocarbon receptor signaling.**
Binge-on-chronic alcohol-fed mice with and without treatment were infected with *Klebsiella* and sacrificed 48 h post-infection. **a** Levels of circulating surfactant protein D1 (SPD-1). **b** Levels of circulating intestinal fatty acid binding protein (IFABP). **c** Number of pulmonary immune cells in binge-on-chronic alcohol-fed mice with and without treatment infected with *Klebsiella*. **d** Number of intestinal immune cells in binge-on-chronic alcohol-fed mice with and without treatment infected with *Klebsiella*. Bars represent the mean ± SEM and dots represent individual mice. *P* values are indicated in the figure or denoted by *$P < 0.05$, **$P < 0.001$, ***$P < 0.0001$, ****$P < 0.00001$, $P < 0.000001$, as determined by one-way ANOVA with Sidak's multiple comparison test. $N = 10$/group.

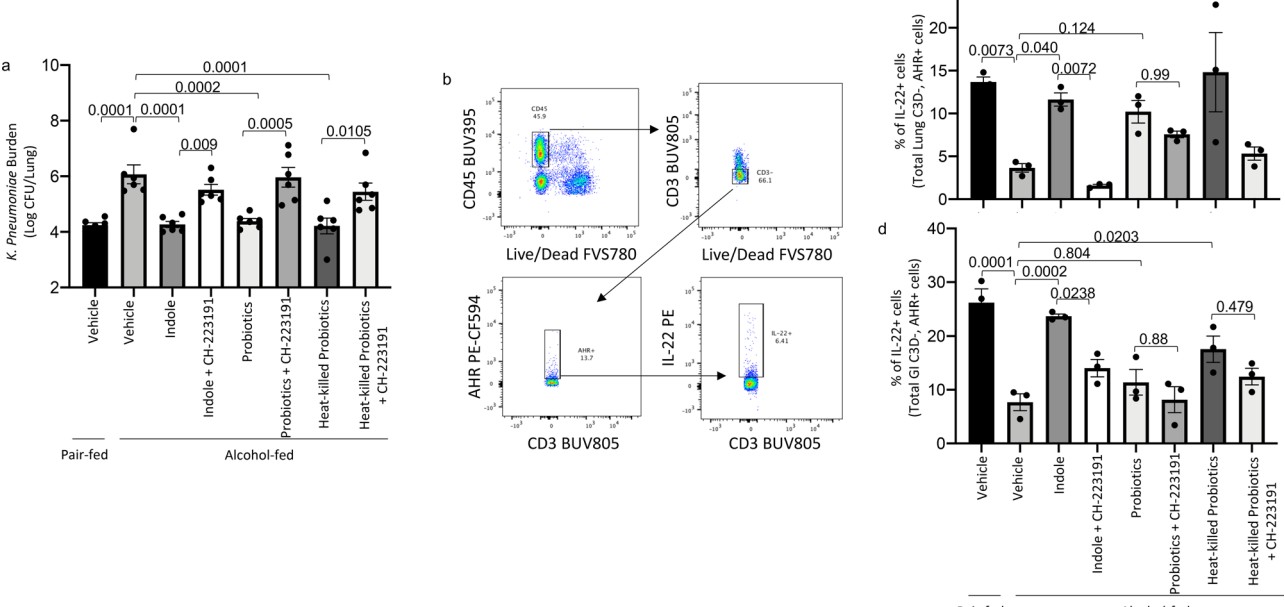

**Fig. 5 Indole and probiotics treatment increase the frequency of IL-22+ immune cells in the lungs and small intestine.** Binge-on-chronic alcohol-fed mice with and without treatment were infected with *Klebsiella* and sacrificed 48 h post-infection. **a** *Klebsiella* lung burden (Log CFU/ml). **b** Flow cytometry gating strategy. **c** Percent of pulmonary IL-22+ immune cells (CD3− AhR+) in binge-on-chronic alcohol-fed mice with and without treatment infected with *Klebsiella*. **d** Percent of intestinal IL-22+ immune cells (CD3− AhR+) in binge-on-chronic alcohol-fed mice with and without treatment infected with *Klebsiella*. Bars represent the mean ± SEM and dots represent individual mice. *P* values are indicated in the figure, as determined by one-way ANOVA with Sidak's multiple comparison test. $N = 3$–6/group.

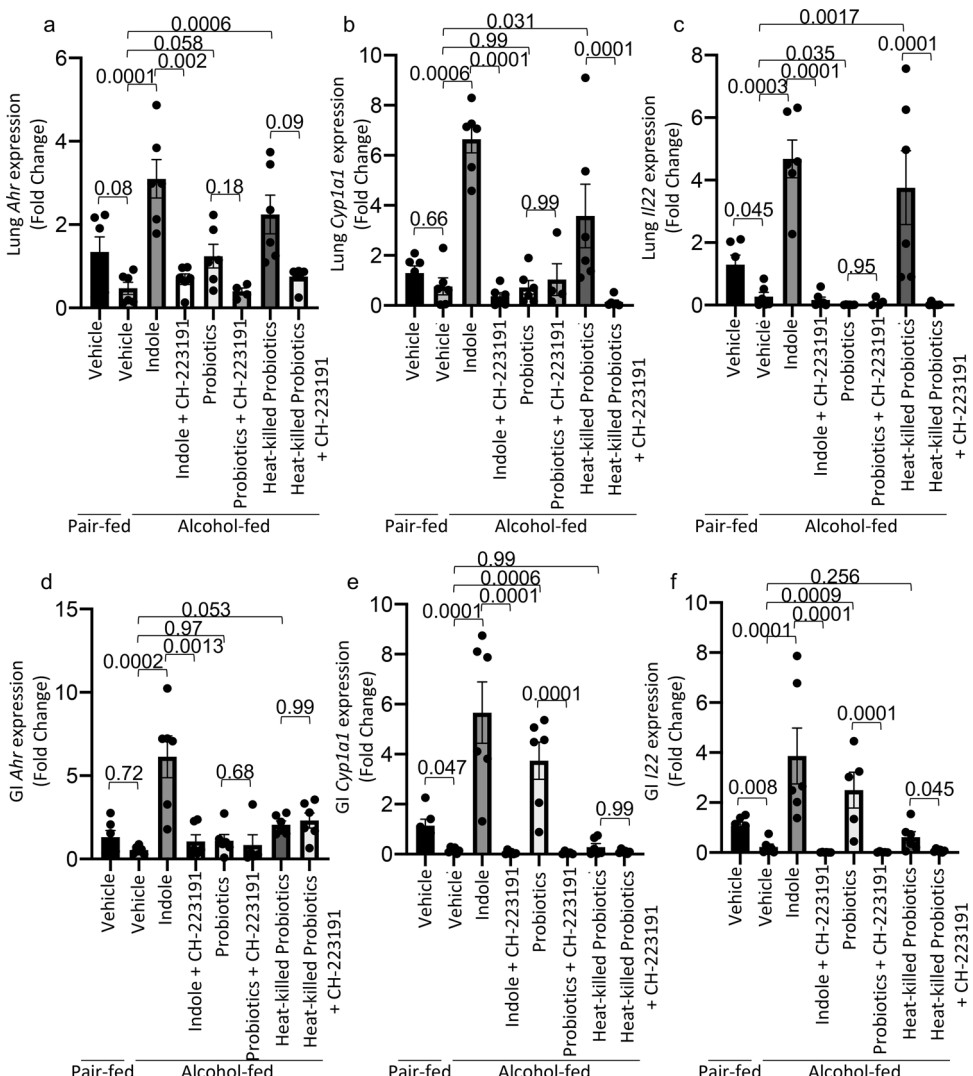

**Fig. 6 Indole and probiotics treatment increase the expression of AhR dependent genes.** Binge-on-chronic alcohol-fed mice with and without treatment were infected with *Klebsiella* and sacrificed after 48 h of infection. Pulmonary expression (fold change) of **a** Ahr, **b** Cyp1a1, and **c** Il22 in binge-on-chronic alcohol-fed mice with and without treatment infected with *Klebsiella*. Small intestinal expression (fold change) of **d** Ahr, **e** Cyp1a1, and **f** Il22 in binge-on-chronic alcohol-fed mice with and without treatment infected with *Klebsiella*. The qPCR value was normalized to GAPDH, and gene expression levels are shown relative to those of control mice. Bars represent the mean ± SEM and dots represent individual mice. *P* values are indicated in the figure and were determined by one-way ANOVA with Sidak's multiple comparison test based off of delta delta CT values. *N* = 6/group.

to control animals following treatment with indole or probiotics (Fig. 6a–f). Further, inhibition of AhR with CH-223191 mitigated the increased expression of *Ahr*, *cyp1a1*, and *Il22* mRNA in treated mice (Fig. 6a–f).

**Microbiota supplementation restores cecal tryptophan catabolite levels.** We then sought to investigate the levels of indole in the cecum, the serum, and the lungs of alcohol-fed mice with and without microbiota supplementation to assess the potential of extra-intestinal indole signaling. Indole levels were decrease in the cecum and the serum of alcohol-fed mice compared to control mice (Fig. 7a, b), while indole levels were below the limit of detection in the lungs of alcohol-fed and control-fed mice (Fig. 7c). Alcohol-fed mice treated with indole had increased levels of indole in all tissues, regardless of treatment with CH223191, which suggests that AhR inhibition does not affect indole levels. (Fig. 7a–c). In fact, indole was only detectable in the lung tissue of animals receiving indole supplementation. In addition, alcohol-fed mice treated with probiotics or heat-killed

probiotics exhibited a non-significant increase in cecal indole levels, and only a modest increase in serum indole levels (Fig. 7a–c). Probiotics did not increase pulmonary indole levels. Suggesting that probiotics likely improve host defense via multiple mechanisms. We also examined the level of several other tryptophan catabolites following treatment with indole or probiotics. The following tryptophan catabolites: indole-3-sulfate, indole-3-acetic acid, and indole-3-lactic acid were decrease in alcohol-fed mice compared to control mice (Supplemental Fig. 3a–d). Treatment with indole restored catabolite levels to control levels, while probiotic treatment restored all catabolites except indole-3-lactic acid (Supplemental Fig. 3d). Indole-3-propionic acid levels were not affected by alcohol-feeding or by the treatments.

**Intestinal microbial products increase AhR activation.** To further evaluate the role of the intestinal microbiota on AhR activation, we utilized an AhR-luciferase reporter assay[38]. Specifically, we utilized the DR-EcoScreen system, which is a mouse

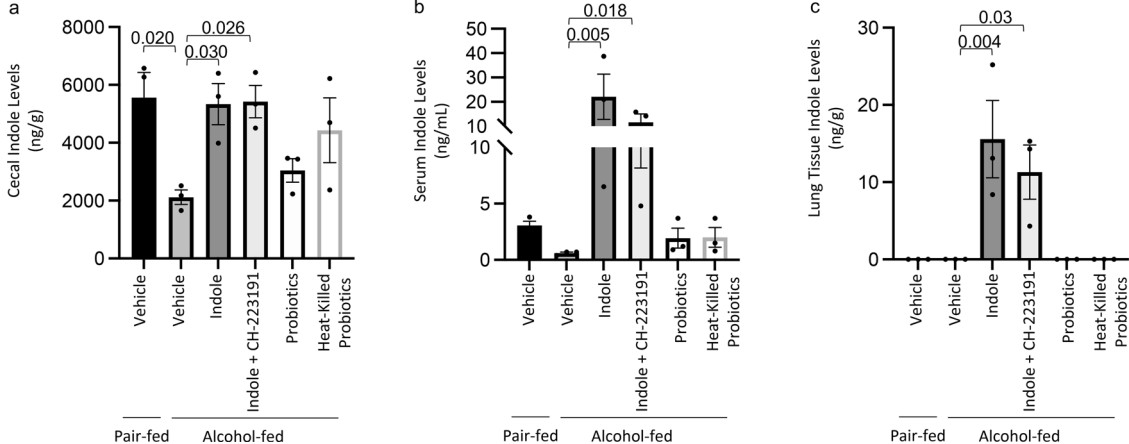

**Fig. 7 Cecal, serum, and lung Indole concentrations.** Binge-on-chronic alcohol-fed mice with and without treatment were infected with *Klebsiella* and sacrificed 48 h post-infection and the indole concentrations in the **a** cecum, **b** serum, and **c** lung tissue were determined. Bars represent the mean ± SEM and dots represent individual mice. *P* values are indicated in the figure and were determined by one-way ANOVA with Sidak's multiple comparison test. $N = 3$/group.

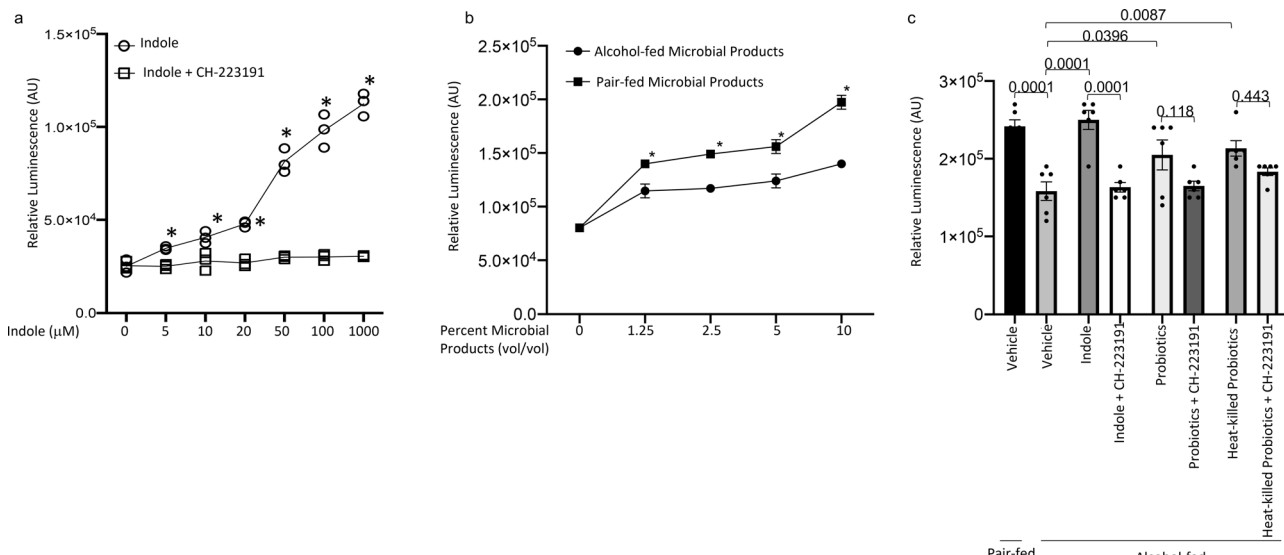

**Fig. 8 Indole and intestinal microbial products from indole and probiotic treated mice increase AhR activation. a** Dose–response curves of indole in the DR-EcoScreen assay. DR-EcoScreen cells were treated with increasing concentrations of indole, to detect AhR agonistic activity. **b** Dose–response curves of intestinal microbial products in the DR-EcoScreen assay. **c** Intestinal microbial products were isolated from binge-on-chronic alcohol-fed mice with and without treatment AhR agonistic activity was determined using the DR-EcoScreen assay. Bars represent the mean ± SEM and dots represent individual mice. *P* values are indicated in the figure and were determined by one-way ANOVA with Sidak's multiple comparison test. $N = 5$–6/group.

derived cellular reporter assay for the ligand-dependent activation of AhR. We first determined the concentration of indole and CH-223191 that activate and inhibit AhR-mediated luciferase production. Indole was added to cells at concentrations ranging from 0 to 1000 μM. In addition, CH-223191 was added to indole treated wells at 30 μM. Indole treatment increase luminescence in a dose dependent manner, starting at 5 μM concentrations when compared to baseline. While treatment of cells with the AhR inhibitor CH-223191 prevented indole-mediated increases in luminescence across all concentrations of indole (Fig. 8a). We likewise confirmed the utility of the DR-EcoScreen system by treating cells with the gold-standard AhR agonist 2,3,7,8-Tetra-chlorodibenzodioxin (TCDD) at concentrations ranging from $10^{-15}$ M to $10^{-10}$ M. Similar to indole treated cells TCDD treatment increase luminescence in a dose dependent manner, and treatment with the AhR inhibitor CH-223191 prevented TCDD-mediated increases in luminescence (Supplemental

Fig. 4a). While murine AhR is activated by indole, TCDD is a much more potent activator of murine AhR. In fact, indole activates AhR-mediated luminescence at ~20% of $10^{-10}$ M (maximum dose) TCDD (Supplemental Fig. 4b). We also sought to determine if the microbial products isolated form the cecum of alcohol-fed and control-fed mice exhibited differential AhR activation. Similarly, we performed a dose response with the cecal microbial products from alcohol-fed and control mice. The microbial products from either ethanol-fed or control mice resulted in activation of AhR above baseline (untreated cells), however microbial products isolated from ethanol-fed mice exhibited a decreased level of AhR activation compared to the microbial products from control mice (Fig. 8b). To further, evaluate the role of the intestinal microbiota we utilized microbial products isolated from all of the treatment conditions and found that similar to in vivo results, the microbial products from indole or probiotic treated mice produced increased AhR activation

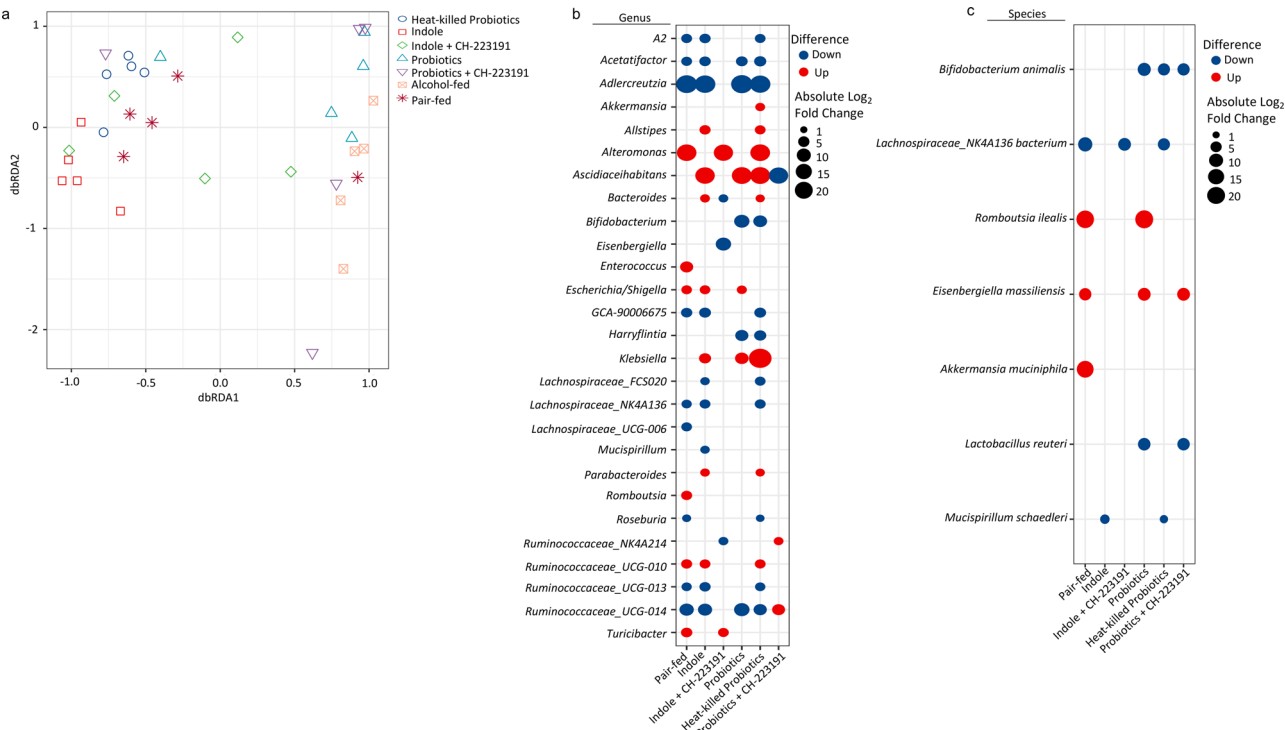

**Fig. 9 Indole and probiotics alter the intestinal microbial structure of alcohol-fed mice.** Binge-on-chronic alcohol-fed mice with and without treatment were infected with *Klebsiella* and sacrificed 48 h post-infection. **a** Beta diversity of alcohol-fed mice with and without treatment, as determined by distance-based redundancy analysis (dbRDA) on sample-wise Bray–Curtis dissimilarity distances. Differentially abundant OTUs at the **b** genus or **c** species level as determined by DESeq2 using a negative binomial generalized linear models for each taxa and Wald test for significances. For each comparison (i.e., columns): taxa with a blue dot (down) indicates that the taxa is decreased in alcohol-fed animals compared to the indicated treatment (i.e., pair-fed, indole, etc.), while taxa with a red dot (up) indicates that the taxa is enriched in alcohol-fed mice compared to the indicated treatment. Absolute indicates the magnitude of the log2fold change. $N = 5$/group.

compared to untreated alcohol-fed mice. Likewise, microbial products isolated from mice treated with the AhR inhibitor exhibited impaired AhR activation (Fig. 8c).

**Microbiota supplementation alters the intestinal microbial communities.** We then assessed alpha and beta diversity of the intestinal microbiota following treatment. No significant differences in alpha diversity were seen among groups. However, beta diversity was significantly altered by alcohol-feeding, as well as by treatment with indole or with probiotics (Fig. 9a). Specifically, beta diversity was different between alcohol and pair-fed mice ($q = 0.0101$), alcohol and indole-treated mice ($q = 0.0144$), alcohol and indole + CH-223191-treated mice ($q = 0.0060$), alcohol and probiotic treated mice ($q = 0.0577$), as well as alcohol and heat-killed probiotic treated mice ($q = 0.0006$). No significant difference was seen between alcohol and probiotic + CH-223191-treated mice ($q = 0.1204$). We then determined the operational taxonomic units (OTUs) that exhibited significantly differential abundance in the cecum at the genus and species levels using DESeq2 (Fig. 9b, c, respectively). Differentially abundant OTUs at the phylum and family levels are shown in Supplemental Fig. 5b, c, respectively. Importantly, *Bifidobacterium animalis* and *Lactobacilus reuteri* (two species present in the probiotic cocktail) were only detected in probiotic treated and the heat-killed probiotic groups (Supplemental Fig. 5a). Indicating that the probiotics are present in the GI tract of the treated mice. Finally, we evaluated the relationship between alpha diversity, beta diversity, and the number of the differentially abundant individual

bacterial genera with *Klebsiella* lung burden. Using general liner model regression, we found that alpha diversity was not significantly associated with *Klebsiella* lung burden ($P = 0.08936$). Similarly, beta diversity was not associated with *Klebsiella* lung burden ($P = 0.3724$). However, beta diversity was associated with feeding and treatments ($P = 0.0001$). Using a negative binomial generalized linear model we found that 7 out of the 27 differentially abundant bacterial genera were associated with *Klebsiella* lung burden. Specifically, the genera *A2* ($q < 0.0001$), *Acetatifactor* ($q = 0.0001$), *Adlercreutzia* ($q < 0.0001$), *Bifidobacterium* ($q < 0.0001$), and *Harryflintia* ($q < 0.0001$) were all negatively correlated with *Klebsiella* lung burden. While the genera *Eisnebergiella* ($q < 0.0001$), and *Klebsiella* ($q = 0.0012$) were positively correlated with *Klebsiella* lung burden.

We also performed PICRUSt2 analysis to assess the predicted functional capacity of the intestinal microbiota. We found that similar to beta diversity, inferred functional capacity was significantly altered by alcohol-feeding, as well as by treatment with indole, probiotics, or heat-killed probiotics, as determined by principle coordinate analysis based off of the inferred functional capacity (Supplemental Fig. 6a). We also evaluated the individual functional pathways of the intestinal microbiota, and found that 132 predicted pathways were significantly different between treatment groups (Supplementary Data 1). A heat-map showing the relative abundance of each predicated pathway in every animal is shown in Supplemental Fig. 5d. Representative functional pathways with significant differential abundance are shown for aromatic amino acid synthesis (Supplemental Fig. 6b), branch-chained amino acid synthesis (Supplemental Fig. 6c), and L-tryptophan synthesis (Supplemental Fig. 6d).

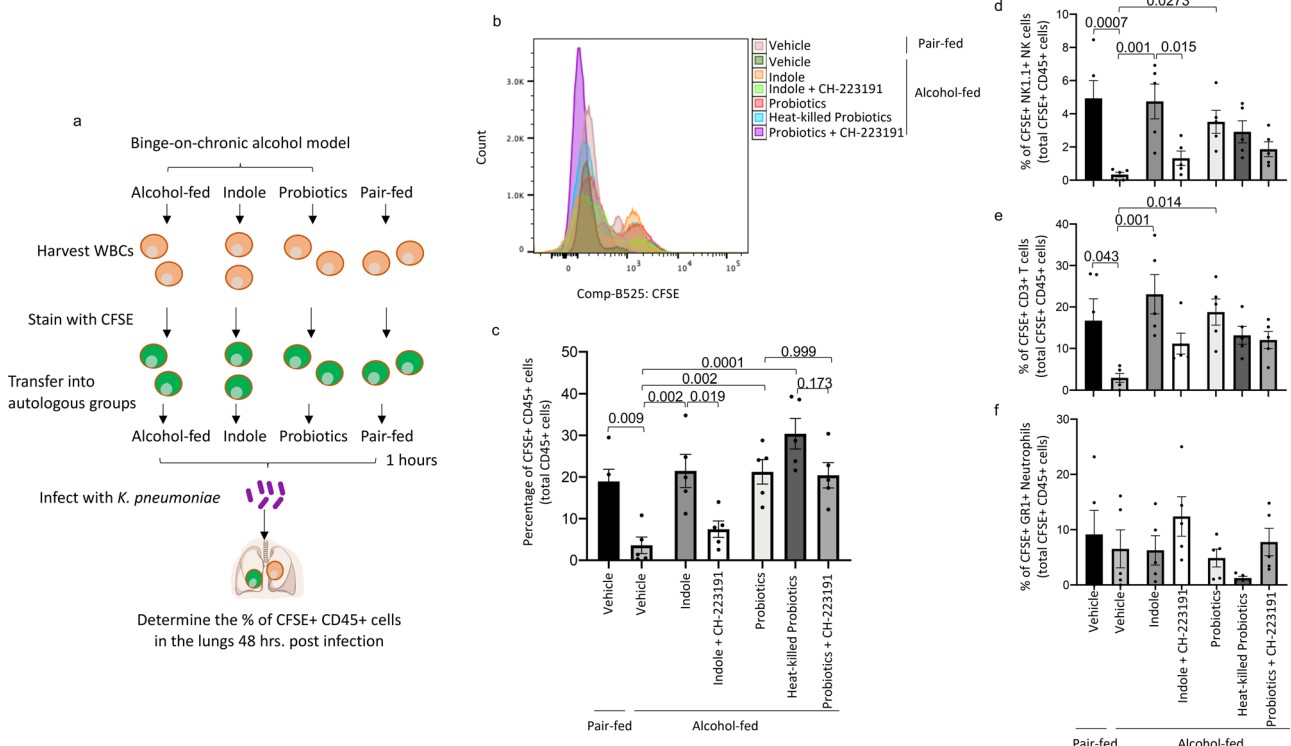

**Fig. 10 Microbiota supplementation mitigates alcohol-associated impairment of immune cell trafficking.** Binge-on-chronic alcohol-fed mice with and without treatment were given CFSE labeled CD45+ immune cells, from autologous treatment donors, 1 h prior to infected with *Klebsiella* and sacrificed 48 h later. **a** Experimental schema. **b** Representative histograms of CFSE+ CD45+ immune cells in the lungs of mice after 48 h of post-infection, all treatments are overlaid. **c** The percentage of CFSE+ CD45+ immune cells in the lungs after 48 h of post-infection. The percentage of CFSE+ CD45+. **d** NK1.1+ natural killer cells. **e** CD3+ T-cells, and **f** GR1+ neutrophils in the lungs after 48 h of post-infection. Bars represent the mean ± SEM and dots represent individual mice. *P* values are indicated in the figure and were determined by one-way ANOVA with Sidak's multiple comparison test. N = 5/group.

**Microbiota supplementation mitigates alcohol-associated impairment of immune cell trafficking**. We then sought to confirm alcohol-induced defects in immune cell trafficking, as the number of lung immune cells were decreased, while the number of intestinal immune cells were increased in alcohol-fed mice in these experiments, as well as our previously published work[31]. Immune cells were harvested from donor mice, labeled with CFSE and used for adoptive transfer into autologous recipients in each treatment group before infection (Fig. 10a). The levels of CFSE labeled immune cells were determined 48 h post-infection. Representative histograms are shown in Fig. 10b. We found that alcohol-fed animals had lower levels of CFSE+ immune cells in the lungs compared to pair-fed mice (Fig. 10c). Further, phenotypic analysis demonstrated that alcohol-fed animals had lower levels of pulmonary CFSE+ NK cells (Fig. 10d), and T-cells (Fig. 10e). However, not difference in levels of pulmonary CFSE + neutrophils (Fig. 10f). Indole treatment, and to a lesser extent probiotics or heat-killed probiotics restored immune cell trafficking to the lungs in alcohol-fed mice (Fig. 10c). Importantly, AhR inhibition prevented the cellular trafficking effects induced by indole, and no effect of AhR inhibition was seen in probiotic treated animals. We then evaluated extra-pulmonary tissues for immune cell recruitment following adoptive transfer. Alcohol-fed mice had higher levels of CFSE+ cells in the intestinal track when compared to control mice (Supplemental Fig. 7a). However, the levels of CFSE+ cells in the liver were reduced compared to control mice (Supplemental Fig. 7b). No differences were seen in levels of CFSE+ cells in the spleen (Supplemental Fig. 7c). Similarly, Indole, probiotics or heat-killed probiotics reversed immune cell trafficking to the intestinal track and the liver in alcohol-fed mice (Supplemental Fig. 7a, b). AhR inhibition prevented the intestinal and liver cellular trafficking effects induced by indole, as well as in probiotic-treated animals. Representative CFSE histograms for each group and tissue is shown in Supplemental Fig. 8.

## Discussion

A plethora of data has emerged demonstrating the importance of the intestinal microbiota for optimal host defense against bacterial and viral respiratory infections[29–31,39–43]. The dynamics of the intestinal microbiota-mediated regulation of the gut-lung axis has been led by work utilizing germ-free and antibiotics-treated mice. For example, germ-free mice, as well as antibiotic treated mice are highly susceptible to pulmonary infection with bacterial pathogens, such as *K. pneumoniae and Streptococcus pneumoniae*[39,40]. Importantly, restoration of the intestinal microbiota via fecal transplant augmented pulmonary host defense, demonstrating that an intact intestinal microbiota is required for optimal pulmonary host defense[39,40].

Several mechanistic pathways have been identified to regulate the gut-lung axis. Brown et al.[40] describe a pathway by which Nod-like receptor-stimulating bacteria in the upper airway and intestinal tract increase levels of interleukin-17A that in turn activates granulocyte–macrophage colony-stimulating factor (GM-CSF) signaling, and stimulates pathogen killing and clearance by alveolar macrophages through extracellular signal regulated kinase signaling[40]. Fagundes et al.[39] describe a pathway by which enhanced susceptibility to *K. pneumoniae* was associated with increased levels of IL-10, which suppresses neutrophil recruitment, and permitted pathogen growth and dissemination. In addition, the administration of LPS prior to pulmonary infection prevented pulmonary *K. pneumoniae* infection, reduced

IL-10 secretion, normalized TNF-α and CXCL1 levels, as well as neutrophil recruitment to the lung[39]. Similarly, we have shown that alcohol-associated dysbiosis, independent of ethanol, increased susceptibility to *Klebsiella* pneumonia[31]. Alcohol-associated dysbiosis was likewise associated with lower immune cell numbers in the lungs, and increased intestinal permeability and inflammation[31]. These studies have several common themes by which the intestinal microbiota influences the pulmonary immune response: (1) regulation of cytokine/chemokine levels both locally (intestinal) and distally (lungs), (2) maintenance of the mobilization of immune cells to the site of infection, and (3) control of the basal activity of pattern recognition receptors.

Taken together, this suggests that the intestinal microbiota is an attractive target for therapeutic intervention for the prevention of dysbiosis-associated pulmonary infections. Most therapeutic approaches utilize probiotics and/or pre/postbiotics to target intestinal dysbiosis. Probiotics have been extensively studied for the treatment or alleviation of many disease, and there is a growing interest in understanding the effect of probiotics on lung disorders such as pneumonia. For example, *Lactobacillus rhamnosus* feeding decreased the burden of *S. pneumoniae* in the lung, prevented dissemination to the blood, and increased INF-γ, IL-6, IL-4, and IL-10 in bronchoalveolar lavage (BAL) fluid in wild-type mice[44]. Interestingly, only one of the two strain of *L. rhamnosus* tested provided a beneficial protective effect against pulmonary infection with *S. pneumoniae*[44]. Similarly, mice administrated *Lactobacillus casei* prior to pulmonary challenge with *Pseudomonas aeruginosa* exhibited increased pathogen clearance, phagocytic activity of alveolar macrophages, and IgA in BAL[45]. Similarly, Hori et al.[46] observed parallel results in murine viral infection models, where feeding mice *L. casei* for 4 months prior to challenge reduced influenza viral titers in nasal washings, and increased natural killer (NK) activity and IFN-γ and TNF-α production[46]. We found marked reductions in the burden of *K. pneumoniae* in the lungs, reduced pulmonary and intestinal permeability, and improved immune response in the lungs of both innate (macrophages, neutrophils, and NK cells) and adaptive immune cells (CD4+ and CD8+ T-cells) in alcohol-fed mice treated with a cocktail of probiotics prior to infection. Interestingly, heat-killed probiotics provided similar levels of protection to live organism. We also see that heat-killed probiotics increase pulmonary IL-22 levels to a greater extent than live probiotics, despite having similar levels of circulating indole. This could be due to several reasons; (1) heat killed probiotic may increase circulating levels of other AhR ligands, other than indole, to a much greater extent than the live probiotics, (2) transcription of the *Il22* gene can also be induced by RORγt[47], and (3) RORγt and AhR can act synergistically to promote IL-22 expression[47]. As such, it is possible that heat-killed probiotics increases RORγt and circulating levels of other AhR ligands leading to a more robust IL-22 response when compared to live probiotics. In similar studies oral feeding of heat-killed *Lactobacillus bulgaricus* was found to attenuate dextran sodium sulfate-induced colitis[48]. Together, these data suggest that active production of specific bacterial metabolites may not be required, but that immune-mediated processing of these beneficial organisms is sufficient.

Short-chain fatty acids (SCFAs) are the most extensively studied bacterial metabolites with immunomodulatory consequences. There is a growing interest in the identification and characterization of specific bacterial metabolites/products with immunomodulatory properties. The catabolism of tryptophan by intestinal commensal organism is one such area of interest, as tryptophan catabolites (indoles) exhibited numerous immunomodulatory effects. Metabolic analyses of human fecal samples detected indole at concentrations of 250–1100 mM[49,50], suggesting tryptophan biotransformation by intestinal microflora is

beneficial for the host. Further, treatment of human intestinal epithelial cells with indole increases barrier function and attenuates inflammatory markers[22]. Additionally, catabolism of tryptophan by *Lactobacilli* hinders colonization of the intestinal tract by pathogenic *Candida albicans* through aryl hydrocarbon receptor (AhR) dependent expression of IL-22[26]. Furthermore, indole, as well as indole-3-acetic acid reduce nonsteroidal anti-inflammatory drug (NSAID) and alcohol-associated liver damage[23,25]. Finally, Tsay et al.[51] showed that antibiotic treatment impaired lung immunity to *P. aeruginosa* by decreasing AhR expression in the intestine and peroxynitrite production by the alveolar macrophages (AMs). They also found that tryptophan or dead *L. plantarum* supplementation decreased *P. aeruginosa* bacterial counts in the lung through increased intestinal ROS production and NF-κB activation, leading to increased phagocytic activity of AMs[51]. We found that alcohol-fed mice treated with a indole prior to infection exhibited marked reductions in the burden of *K. pneumoniae* in the lungs, reduced pulmonary and intestinal permeability, and improved immune response in the lungs of both innate (macrophages, neutrophils, and NK cells) and adaptive immune cells (CD4+ and CD8+ T-cells). Indole's protective effects were also dependent on AhR signaling, as inhibition of AhR by CH-223191 mitigated indole's immunomodulatory effects. Similarly, we found that indole treatment rescued immune cell trafficking to the lungs following *K. pneumoniae* infection. It is important to note that indole preferentially activates human AhR (EC$_{50}$ ~ 3 μM) compared to murine AhR[52]. However, our results along with other published data[26,51–53] shows that mouse AhR is activated by indole, but that TCDD is a much more potent activator of murine AhR. We and others hypothesis that TCDD causes a sustained and inappropriate activation of AhR, which leads to many detrimental effects such as clonal expansion of preneoplastic cells, or inhibition of apoptosis. In contrast, indole induces transient lower (than TCDD) activation of AhR, which we believe to be critical for maintaining normal cellular homeostasis. Clearly, additional research is needed to understand the balance between beneficial and detrimental effects of AhR signaling. Furthermore, research is needed to understand if the positive and negative effects of AhR signaling are either ligand and/or dose-dependent.

Taken together our results, along with the published literature, suggest that indole treatment restores: (1) intestinal barrier function; (2) cytokine/chemokine levels; (3) immune cell activation and programing; (4) immune cell recruitment to infected tissues; and (5) protection against bacterial pneumonia. These data also suggest that manipulation of tryptophan catabolism, AhR signaling, or IL-22 production during disease both locally (intestinal) or distally (lungs) should be further explored as novel therapeutic approaches for the prevention of alcohol-associated pneumonia. Interestingly, nearly every study to date that has utilize either probiotics and/or bacterial metabolites employs a prophylactic treatment approach for the mitigation or prevention of the disease of interest. However, as these therapies move into the clinical realm it will be important to understand if probiotics and/or metabolites can also be used as therapeutic interventions.

Treatment of alcohol-fed mice with the probiotic cocktail, heat-killed probiotics, or with indole lead to a reduction in pulmonary bacterial burden. Suggesting that alcohol-mediated changes to the intestinal microbiota is a critical mechanistic pathway by which alcohol impairs pulmonary host defense against bacterial pneumonia. However, there were specific effects on immune cell profiles, the intestinal microbial communities, as well as the involvement of AhR signaling unique to each treatment condition. For example, AhR inhibition leads to mitigation of every beneficial effect following indole treatment, suggesting that indole mainly acts as an AhR agonist. However, AhR inhibition does not

mitigate all the beneficial effects of probiotics or heat-killed probiotics treatments. Specifically, AhR inhibition does not impair the trafficking of immune cells to the lung in response to pneumonia in animals that received probiotic supplementation. While AhR inhibition does blunt the effects of probiotics on bacterial burden, immune cell numbers, epithelial permeability, and gene expression there is often no statistical difference between probiotic treated mice and probiotic-treated mice given CH-223191, while AhR inhibition in indole-treated mice is always significantly different from indole treated alone. This strongly suggests that probiotics or heat-killed probiotics act via multiple pathways. However, the lack of an effects of probiotics on AhR signaling may also be due to several reasons. First, microbial tryptophan catabolites have been shown to have both agonistic and antagonistic effects against other microbial and environmental AhR ligands[54]. For example, indole-3-acetamide and indole-3-pyruvate exhibit dual effects on ligand-activated AhR. Each of the indole metabolites antagonize TCDD but potentiate the agonistic effects of BaP and FICZ[54]. Given this data, it is possible that probiotic treatments results in a shift in the production and/or amount of different indole metabolites, such that one metabolite acts as an AhR antagonistic in our model system. This may help to explain the intermediate effects of CH-223191 treatment that we observed in the probiotic treated mice. Finally, the kinetics of indole production and/or conversion may be different between probiotic-treated mice and indole-treated mice, as we only evaluated mice at one time point post infection. As such, it is possible that the probiotic-mediated effects on AhR signaling occur early then we examined. These data strongly support the need for a more complete understanding of the link between probiotics, indole, and AhR signaling in the context of host-defense against alcohol-associated pneumonia.

The results from this study demonstrate that intestinal microbiota targeted therapeutics may be used to mitigate the risk of alcohol-associated bacterial pneumonia. Specifically, treatment with the specific microbial metabolite indole or with a probiotic cocktail may mitigated alcohol-associated increase in pathogen burden, restored pulmonary and intestinal immune responses, as well as pulmonary immune cell trafficking. Protective effects are, in part, mediated by AhR, as inhibition of AhR diminished the protective effects of indole. These findings suggest that alcohol either directly or through its metabolism, impairs the production/processing of microbial metabolites, specifically tryptophan catabolites, resulting in immune dysregulation and impaired cellular trafficking required for optimal host defense.

## Methods

**Mice.** Female 10–12-week-old C57BL/6 mice were obtained from Charles Rivers Breeding Laboratories (Wilmington, MA) and maintained in a temperature-controlled room for a week prior to experimental manipulation. Animals were kept in the animal care facility at LSUHSC throughout the experiment. Animals were handled under a laminar flow hood to maintain SPF conditions throughout the course of the experiment. All experiments were approved by the Institutional for Animal Care and Use Committee (IACUC) at LSUHSC.

**Alcohol feeding model.** We utilized a binge-on-chronic alcohol feeding model[31]. Briefly, mice were acclimated to liquid diet for 5 days using Lieber-DeCarli '82 Shake and Pour (Bioserv, Flemington, NJ). Groups of mice ($N = 5$ per cage) were randomized into ethanol-fed, ethanol-fed plus treatment, or pair-fed groups. Pair-fed were maintained on control-liquid diet adjusted daily according to the consumption of ethanol-fed mice. Mice were administered 4 g kg⁻¹ (31.5% vol/vol) ethanol by gavage following 5 and 10 days of alcohol-feeding. Pair-fed control mice were gavaged with 7 g kg⁻¹ (45% wt/vol) maltose dextrin.

**Treatments.** All mice were treated by oral gavage daily thorough out the course of the experiments. Mice received indole by gavage (20 mg/kg/dose or ~200 μL of 17 mM indole; Sigma Aldrich, St. Louis, MO) dissolved in sterile water warmed to 55 °C. All probiotics (*Bifidobacterium bifidum, Bifidobacterium lactis, Lactobacillus plantarum, Lactobacillus reuteri,* and *Lactobacillus rhamnosus*) were obtained from

DuPont™ Danisco® FloraFIT® probiotics as lyophilized powder at a known CFU/gram concentration. Each probiotic was weight and aliquoted into $1 \times 10^8$ CFU aliquots. Probiotics or heat-killed probiotics were resuspended in PBS and mice were gavaged with $5 \times 10^8$ CFU/dose. Probiotic cocktails were heat-killed by incubating at 100 °C for 15 min, bacterial viability was then confirmed by plating. No viable organisms were recovered from the heat-killed probiotics. Mice received CH-223191 by gavage (10 mg/kg/dose or ~200 μL of 3 mM CH-223191; Sigma Aldrich) dissolved in sterile corn oil. Alcohol-fed and pair-fed mice were gavage daily with vehicle (PBS). A separated cohort of control mice was given corn-oil or water daily to ensure no effects were seen with the different vehicles. No changes in host defense or bacterial burden were seen in control mice given corn-oil or water.

**Klebsiella pneumoniae infection.** *Klebsiella pneumoniae* infections and burden assessments were performed using standard methods[31]. Briefly, *K. pneumoniae* (strain 43816, serotype 2; American Type Culture Collection, Manassas, VA) was grown in 100 mL tryptic soy broth (Becton Dickinson, Franklin Lakes, NJ) in a shaking incubator (185 r.p.m.) at 37 °C for 18 h. Bacteria were then pelleted by centrifugation ($2000 \times g$ for 15 min at 4 °C), washed twice with phosphate-buffered saline (PBS), and resuspended in PBS at an estimated concentration of $1 \times 10^3$ colony-forming units (CFU)/mL. Recipient mice were lightly anesthetized with isoflurane (1–4% to effect). Animals were suspended by their front incisors, the tongue was gently extended out with forceps, and 100 μL inoculum ($1 \times 10^2$ CFU) was injected into the trachea using a P200 pipette. After inhalation of inoculum was observed, the tongue was released, and the animal was allowed to recover from anesthesia. All mice were sacrificed 48 h post-infection, and pulmonary and splenic burden was determined via serial dilution and plating onto HiCrome Klebsiella Selective Agar plates.

**SPD-1 and IFABP ELISA.** Serum was collected from all mice at sacrifice using BD serum separator tubes (BD Biosciences, San Jose, CA). Serum was then used for SPD-1 and IFABP ELISAs according to manufacturer's specifications (R&D systems, Minneapolis, MN).

**Flow cytometry.** Lungs and intestinal immune cells were collected for flow cytometry analysis[31]. Lung and intestinal tissue of each animal was minced, suspended in 10 mL homogenization buffer consisting of RPMI 1640 medium with 1 mg/mL Collagenase type 1 (Worthington Biochemical, Lakewood, NJ) and 30 mg/mL DNase I (Roche Diagnostics, Indianapolis, IN), and incubated at 37 °C with shaking for 30 min. Cell suspensions were further disrupted by passing through a 70-mm nylon mesh. Intestinal lymphocytes were further purified by 44/67% Percoll gradient. RBCs were lysed using RBC lysis buffer (BioLegend, San Diego, CA) prior to staining. After washing with PBS, viable cells were counted on a hemocytometer using the trypan blue–exclusion method. One million viable cells were then used for staining. Both lung and intestinal immune cells were pretreated with TruStain FcX Anti-mouse CD16/32 antibody (2 μL per test, Cat #101320, BioLegend, San Diego, CA) and then stained with eFlour780 fixable viability dye (1 μL/mL, Cat #65-0865-14, Invitrogen, Eugene, OR) or FVS780 (1 μL/mL, Cat #565388, BD) followed by staining with antibodies specific for murine CD45 (0.25 μg per test, Cat #103104, BioLegend), CD3e (2.5 μg per test, Cat #100216, BioLegend), CD4 (1 μg per test, Cat #564667, BD), CD8a (1 μg per test, Cat #100748, BioLegend), GR1 (0.25 μg per test, Cat #108452, BioLegend), NK1.1 (2.5 μg per test, Cat #108720, BioLegend), CD11c (1 μg per test, Cat #117328, BioLegend), CD11b (0.4 μg per test, Cat #101263, BioLegend), F4/80 (0.5 μg per test, Cat #123149, BioLegend), CD45 (0.25 μg per test, Cat #565967, BD), CD3e (0.5 μg per test, Cat #749276, BD), AhR (0.5 μg per test, Cat #565790, BD), and IL-22 (0.125 μg per test, Cat #1H8PWSR, Thermo). Streptavidin Qdot 585 (10 nm per test, Cat #Q10111MP, ThermoFisher) was used in conjunction with biotin labeled antibodies. For all experiments, cells were acquired using an LSR II flow cytometer (BD Biosciences). Cell numbers were determined using Precision Count Beads according to manufacturer's instructions (Cat #424902, BioLegend).

**DR-EcoScreen cells and AhR activation.** DR-EcoScreen cells were obtained from Sekisui XenoTech via the JCRB Cell Bank at the National Institute of Biomedical Innovation, Health and Nutrition. Ligand-dependent activation of AhR was determined using the mouse derived DR-EcoScreen cell line, as described previously[38]. Briefly, cells were trypsinized and suspended at a density of $1.0 \times 10^5$ cells/mL in a-MEM containing 5% of FBS. The cell suspension was seeded in a well of a 96-well white bottom plate at a final density of 50,000 cells/well (Nunc, Denmark). Various concentrations of chemicals (indole, TCDD, or CH-223191) that were dissolved in 1% DMSO was added to each well (final concentration of DMSO was 0.1%) and incubated for 24 h at 37 °C in an atmosphere of 5% $CO_2$/95% air under saturating humidity. Following 24 h of incubation, Steady-Glo™ reagent was added to each well and the plate was shaken at room temperature for 5 min, the luminescence was then measured with a microplate-luminometer.

**Quantitative RT-PCR.** Intestinal and lung samples were harvested at sacrifice and placed directly into Allprotect Tissue Reagent (Qiagen). Total RNA was extracted using the RNeasy Plus Mini Kit on the QIAcube Connect (Qiagen). CYP1A1, AhR, and IL22 mRNA levels in the lungs and small intestine of treated mice treated was measured by real-time quantitative PCR, using QuantiTect SYBR® Green RT-PCR

Kit (Qiagen), according to manufactures specifications. The following primer sequences were obtained from the PrimerBank database[55,56]: CYP1A1 (PrimerBank ID: B6753564a1), AhR (PrimerBank ID: 19526637a1), IL-22 (PrimerBank ID: 21426819a1), and GAPDH (PrimerBank ID: 6679939a1). The amounts of CYP1A1, AhR, and IL22 mRNA were normalized to that of GAPDH mRNA.

**Immune cell isolation and CFSE**. Spleen were harvested at sacrifice, resuspended in PBS, and homogenized using gentleMACS™ Octo Dissociator (Miltenyi Biotec, Auburn, CA). Whole blood was collected using BD Microtainer® Blood Collection Tubes with K2EDTA (BD Biosciences). Splenic homogenates and whole blood were then combined and subjected to 1-Step Polymorphs density gradient centrifugation (Accurate Chemical, Westbury, NY). Both PBMCs and PMN buffy coats were harvested and combined. Immune cells were then stained using Cell-Trace™ CFSE Cell Proliferation Kit according to manufacturer's specifications (ThermoFisher). Viable cells were counted and ~$1 \times 10^7$ immune cells were given to each recipient mouse via IP injection.

**Liquid chromatographic and mass spectrometric conditions for indole quantitation**. Mass spectrometric detection was performed on an LC-MS/MS 8060 system coupled with a Shimadzu Nexera UPLC system (Shimadzu Scientific Instruments, Columbia, MD). All chromatographic separations were performed with a Synergi Fusion-RP C18 column (4 u; 250 × 2.0 mm) equipped with a Phenomenex C18 guard column (Phenomenex, Torrance CA). The mobile phase consisted of 0.1% formic acid in water (mobile phase A) and methanol (mobile phase B) operated at total flow rate of 0.25 mL/min. The chromatographic separation was achieved using 12 min gradient elution with mobile phase composition changing from 50% B at 2 min to 85% B at 5 min. Ionization was carried out using atmospheric pressure chemical ionization (APCI) operated in positive mode. Multiple reaction monitoring (MRM) mode was used for quantitation using precursor ion > product ion combinations of 118.15 > 91.1 $m/z$ for indole, and 124.15 → 98.1 $m/z$ for indole-d7 used as an internal standard (IS). The assay was linear over the range of 1–500 ng/mL. Indole was extracted from calibration standards, quality control, serum, lungs, and cecum samples by protein precipitation using acetonitrile.

**DNA sequencing of the 16s rRNA gene**. Sequencing was performed by the Louisiana State University School of Medicine Microbial Genomics Resource Group (http://metagenomics.lsuhsc.edu/mgrg)[31,57]. Briefly, cecal contents were flash frozen and genomic DNA extraction was performed using the QIAamp DNA StoolMini Kit (Qiagen Valencia, CA, USA) modified to include bead-beating. The 16S ribosomal DNA hypervariable regions V3 and V4 were amplified using gene-specific sequences, Illumina adapters, and molecular barcodes primers. Precisely, the V3 Forward: 5′- CCTACGGGAGGCAGCAG-3′ with a forward adapter (5′-AATGATACGGCGACCACCGAGATCTACAC-3′), a forward pad: (TATGGT AATT), and a forward linker (GG), and the V4 Reverse: 5′-GGACTACHVGGG TWTCTAAT-3′ with a reverse adapter (5′-CAAGCAGAAGACGGCATACGAGA T-3′), a reverse pad (AGTCAGTCAG), and a reverse linker (CC) were used for sequencing of cecal DNA samples. Samples were then sequenced on an Illumina MiSeq (Illumina, San Diego, CA, USA) using the 2 × 300 bp V3 sequencing kit. In addition, internal sequencing controls (negative water control and a positive mock community control) were included to test for contamination of the sequencing runs.

**Sequence analysis**. Raw sequence data was processed using R and the R packages: DADA2 v1.1.5, Phyloseq v1.16.2, DESeq2 v1.20.0, PICRUSt2 v2.2.0, STAMP v2.1.3, and vegan v2.3-5[58–64]. Sequences were truncated, denoised, chimera-filtered, and clustered into sequence variants using DADA2. Operational taxonomic units (OTU) were generated in DADA2 by taxonomic classification of sequence variants using the SILVA reference database v132. The number of unique sequence variants in a sample (α-diversity) was calculated using the estimate_richness function in Phyloseq. Beta-diversity analysis was performed using a distance-based redundancy analysis (dbRDA) on sample-wise Bray–Curtis dissimilarity distances using Phyloseq and vegan. Differentially abundant OTUs were determined via DESeq2. Inferred functional capacities were determined using PICRUSt2 and statistical significance was determined using STAMP. Graphics were generated using ggplot2 and STAMP.

**Statistics and reproducibility**. Statistical analyses were performed using Graph-Pad Prism 8 (La Jolla, CA) and the R package vegan. Results are shown as the mean ± standard error of the mean. A $P < 0.05$ and a false discovery rate (FDR) $q$-value < 0.1, were deemed significant. Sample size and number of replicates is indicated in each respective figure legend. Statistical significance was assessed using a Mann–Whitney test for comparisons between two groups and a one-way analysis of variance (ANOVA) with Sidak's multiple comparison test for comparisons between three or more groups. The statistical significance of the different microbiome measurements was assessed as follows. Alpha-diversity significance was inferred using Wilcoxon tests on vectors of data with corrections for multiple

comparison via FDR. Beta-diversity significance was inferred via permutational multivariate analysis of variance using distance matrices with corrections for multiple comparison via FDR. Assessment of the relationship between *Klebsiella* burden and alpha and beta diversity was done using a generalized linear model (GLM) and adonis2, respectively. Assessment of the relationship between *Klebsiella* burden and the differentially expressed bacterial genera was done using a negative binomial generalized linear model with corrections for multiple comparison via FDR. Model variables included *Klebsiella* burden, alpha or beta diversity, treatment group, and sequencing depth. Differentially abundant OTUs were determined using a negative binomial distribution model from DESeq2. Inferred functional capacities were determined using PICRUSt2 and statistical significance was determined using ANOVA followed by Tukey-Kramer post-hoc analysis with corrections for multiple comparison via FDR within the STAMP platform.

**Reporting summary**. Further information on research design is available in the Nature Research Reporting Summary linked to this article.

## Data availability
Sequencing data is deposited in the National Center for Biotechnology Information Sequence Read Archive (BioProject ID: PRJNA602539). All additional data, including mass spectrometry-based indole quantification, which support the findings of this study are available within the paper and its supplementary information files (Supplementary Information and Supplementary Data 1).

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

## Acknowledgements

We would like to especially thank Daryl J. Murry, Yashpal S. Chhonker and Vineet A. Joshi for their help with indole quantification via mass spectrometry. This work was supported by the National Institute of General Medical Sciences of the National Institutes of Health, which funds the Louisiana Clinical and Translational Science Center, Grant #U54-GM104940, and by the National Institute on Alcohol Abuse and Alcoholism Grants #P60-AA009803, #K99-AA026336, #R21-AA027199, and #R00-AA026336. The content is solely the responsibility of the authors and does not necessarily represent the official views of the National Institutes of Health. The funders had no role in study design, data collection and analysis, decision to publish, or preparation of the manuscript.

## Author contributions

Conception: D.R.S. and D.A.W.; research design: D.R.S., M.G., J.E.S., P.E.M. and D.A.W.; investigation: D.R.S., M.G., C.M.T. and M.L.; data analysis: D.R.S., M.G., C.M.T. and M.L.; writing—draft and editing: D.R.S., M.G., J.E.S., P.E.M., C.M.T., M.L. and D.A.W.; and project administration: D.R.S.

## Competing interests

The authors declare no competing interests.
