## [Peer Review File · Communications Biology]

Reviewers' comments:

Reviewer #1 (Remarks to the Author):

COMMSBIO-20-1037-T

The paper by Samuelson et al investigates how intestinal microbial dysbiosis induced by alcohol, influences susceptibility of mice to infection with *Klebsiella pneumoniae*. The paper is a follow up to their previous two studies. The authors show that chronic alcohol administration decreases levels of some tryptophan microbial catabolites, including indole, which in turn results in increased susceptibility of mice to KP infection. Administration of mice with indole or probiotic cocktail reversed alcohol effects on KP infection rate and also mitigated some inflammatory parameters. The involvement of aryl hydrocarbon receptor AhR in the process is anticipated, because addition of AhR antagonist CH223191 abrogated the attenuation of infection and inflammation by indole. Overall, the study is rigorously elaborated; however, there is a number of specific conceptual issues that significantly dampen the enthusiasm for the paper.

1/ There is no mention in abstract and introduction, on the roles on tryptophan intestinal microbial catabolites in physiology and patho-physiology, but the paper is focused on these compounds.

2/ The authors say (in abstract), that alcohol impairs the production of microbial catabolites, "specifically indole". However, in Fig 5, they show that alcohol also drastically diminishes the levels of I3-sulphate, I3-acetate and I3-lactate.

3/ There are concerns about intestinal/fecal concentration of indole, reported in the study. P12 (discussion) the authors cite paper, where are reported indole fecal concentration in humans about 250-1100 nM. Cecal concentrations of indole (Fig 5) were determined to be around 10 nmol/g (i.e. approx. 10 μ M). Mice were administered with 20 mg/kg/dose of indole. It should be clarified how are related these three concentrations, and also using common units to allow comparison is recommended.

4/ Indole was described as human-specific ligand of AhR (Hubbard et al 2015 Sci Rep). An affinity of indole against mouse AhR was recently reported, but IC50 in competitive ligand-binding assay was higher than 1 μ M. Also EC50 in reporter assays in human cells by indole was about 1.5 μ M (Vyhliadalova et al 2020 Int J Mol Sci). These concentrations, both in human and mice models, are apparently much higher than those used in the study by Samuelson et al. How to explain? Fig 5A – indole cecal levels are restored in indole-treated mice back to approx. 10 μ M, hence too low to activate AhR.

5/ The involvement of AhR in the effects observed by indole is not convincingly evidenced in the paper: (i) There is absent any proof of AhR activity (basal, induced, inhibited), e.g. by AhR target gene expression etc.; (ii) The use of AhR antagonist CH223191 is inappropriate and insufficient approach to demonstrate AhR involvement. This antagonist is ligand selective, e.g. it antagonize effects by TCDD but not by benzo-a-pyrene. Hence, dose-response effects of indole (itself and/or combined with CH223191), either in cell-based assay, or in vivo gene expression should be shown.

6/ Conceptual: There is presented that alcohol decreases indole levels in cecum, and indole supplementation restores indole levels. It is kind of tautology, when showing that a drop in concentration of the compound is reversed by addition of the same compound. There should be demonstrated clear link between plasma and intestinal/fecal levels of indole in control and alcohol-fed mice and indole-fed mice. It is essential to distinguish between mechanistic base of indole effects, or simple external replenishment of depleted indole levels.

Reviewer #2 (Remarks to the Author):

1. Fig 1b, 1c, 3b, 3c: There are a number of data points along the x axis. Did these animals have undetectable *K. pneumoniae* burden? Does the extent of colonization of the probiotic relate to the amount of *K. pneumoniae* burden, i.e., those animals with a higher lung burden had a corresponding lack of colonization by the administered probiotics?
2. The authors should quantify intestinal and pulmonary IL-22 production to confirm active AhR signaling.
3. In figure 6c, I do not see any Lactobacilli in the probiotic-treated groups. The probiotic mixture contains 2-3 Lactobacilli, so can the authors comment on why not detected? Can they use 16S qPCR to quantify whether these species engrafted at all (also relevant for point #1).
4. For figures 6d-f, it is surprising that no inferred pathways were significantly changed with live probiotics, whereas this change is observed with heat-killed probiotics. Can the authors comment? Moreover, it appears that CH-223191 (i.e. AhR inhibition) does not alter any of these inferred bacterial pathways either, which presumably would be impacted by altered host AhR signaling.
5. Heat killed probiotics also have protective effects, which suggests (also mentioned by authors) an additional mechanism of protection. Can the authors comment on what it could be? Could it also be dependent on AhR signaling? While not necessary for this manuscript, the authors should consider a future study in which heat-killed probiotics administered arm is also treated with CH223191.
6. These studies indicate that prophylactic probiotics and indole administration (and AhR activation) can protect from binge-drinking associated *K. pneumoniae* pathogenesis. However, can these be used as therapeutic interventions? Here, comments + citations are sufficient.
7. Minor:
 - v In the Abstract line 34, I think the authors mean "mitigate" and not "migrate".
 - v Please describe how bacteria were heat killed.
 - v Make data points in the PcoA plot in 6b larger as they're hard to see.
 - v Please check figure legends for lower case p in pneumoniae.

Reviewer #3 (Remarks to the Author):

This study investigates the impact of indole and probiotic treatment on *K. pneumoniae* lung infection in alcohol-fed mice. Previously, the authors had shown that alcohol consumption associated dysbiosis increases the susceptibility of mice to *K. pneumoniae* infection in the lungs. Overall, the results shown here support the premise that raising the levels of microbiota metabolites, specifically tryptophan catabolites in the intestine, could improve immune cell counts in the lung while reducing *K. pneumoniae* burden. Using a chemical inhibitor, the authors conclude that the effects of indole are mediated by the AhR. This conclusion requires additional, evidence, as the authors do not provide direct evidence of AhR activation by indole in target tissues. Further, several aspects of the study require further explanation. The discussion could also be improved by tying together the observations regarding indole and probiotic treatment. Some conclusions, e.g., cytokine modulation, appear to be speculative, as the paper does not present these data. Specific comments follow.

1. Please comment on the distribution of responses between animals. It seems that some animals in the treatment groups (indole, probiotic) had zero *K. pneumoniae* burden in both the lung and spleen, and this is what decreased the group averages. In treated animals that showed *K. pneumoniae* burden, the CFUs seem comparable to the alcohol-fed controls.

2. In addition to IFABP, have the authors measured LPS in circulation? Translocation of bacterial endotoxin would more directly indicate a weakened barrier function in the context of an inflammatory response. Some studies, e.g., Lau et al., (2016) have suggested that IFABP is not a reliable marker of intestinal barrier dysfunction. Further, IFBAP is expressed in epithelial cells of the small intestine, whereas the microbiota analyses in this paper focused on the cecum.

3. I have several concerns regarding the AhR experiments. What concentration of indole in the various tissues, i.e., lung, intestine, spleen, etc.? What is the evidence that indole activated the AhR as a ligand? Have the authors measured a panel of AhR genes, e.g., CYPs, in the relevant tissues? The text implies that IL-22 production should be modulated. Have the authors measured the effect of indole on IL-22 production in intestinal immune cells (e.g., Th17 cells)? Along this vein, have the authors checked that CH-223191 is detected at sufficiently high (inhibitory) concentrations in the target tissues and that expression of AhR genes is down-regulated?

4. The connection between CH-223191 and probiotic treatment is unclear. The authors point out that live and heat-killed probiotics elicited similar responses, suggesting that metabolic activity of the probiotics is not necessary. Does this mean that the heat-killed probiotics contained AhR ligands?

5. Results shown in figure 5 indicate that alcohol feeding depletes several tryptophan catabolites in the cecum, which are restored if the treated animals receive indole or a probiotic cocktail. Given the above question, it would be interesting to know if the heat-killed probiotic also raised the levels of the metabolites. What are the levels of these metabolites in the lung and spleen?

6. I found the description of the microbiome sequencing results difficult to follow. Figure 6A includes a symbol (plus) that does not map to any experimental group. These plots show N = 5, rather than N = 10. It would be interesting to know if the treated animals in this figure correspond to the animals that showed minimal (~zero). The probiotic group in Figure 6A projects closer to the alcohol-fed group than to the pair-fed group. However, the text states that the beta diversity is significantly different between these two groups. Curiously, the heat-killed probiotic group projects closer to the indole and pair-fed group than the live probiotic group.

7. The PCA scatter plot shown in Figure 6B is very difficult to read. The symbols are small and have similar colors. What is meant by "inferred functional capacity?"

8. Figure 6C indicates that the relative abundance of the genus *Klebsiella* is substantially higher in several treatment groups compared to the pairwise-fed control, including indole, probiotic, and heat killed probiotic. Does this genus abundance count exclude the pathogen? Further, why would the *Bifidobacterium* count decrease in probiotic treated animals when the cocktail includes this genus? The Results section makes no mention of these observations. Also, given the large number of changes shown in the figure, the following statement is confusing: "In most instances treatment with indole, probiotics, or heat-killed probiotics restored the relative abundance of OTUs to levels similar to those seen in pair-fed control mice ..."

9. Have the authors measured representative metabolites from the pathways shown in Figures 6D-F, e.g., tyrosine, phenylalanine, tryptophan, valine, etc.? Having the metabolite data could corroborate the PICRUSt2 results.

10. If neither microbiome measure (alpha or beta diversity) is a significant predictor of *K. pneumoniae* burden, then what is the link in the gut-lung axis the authors describe?

11. I would have liked to see a discussion of the different effects of probiotic and indole treatments. The AhR inhibitor experiments suggest that CH-223191 attenuated some of the effects of probiotic treatment, e.g., SPD-1, immune cell counts (Figure 4). This suggests that there could be a shared mechanism, but the two treatments are discussed separately.

12. Additional controls are needed to establish that indole is acting through the AhR (see point #3). The possibility that indole treatment restores cytokine/chemokine levels is intriguing, but this seems to be speculative, as these data are not part of this study. Similarly, the authors did not directly assess if indole treatment impacted immune cell programming.

13. The timing of treatment is interesting. What would happen if the animals were exposed to indole and/or probiotic post-infection rather than pre-infection (which would more closely mimic the administration of a therapeutic)?

Reviewers' comments:

Reviewer #1 (Remarks to the Author):

COMMSBIO-20-1037-T

The paper by Samuelson et al investigates how intestinal microbial dysbiosis induced by alcohol, influences susceptibility of mice to infection with *Klebsiella pneumoniae*. The paper is a follow up to their previous two studies. The authors show that chronic alcohol administration decreases levels of some tryptophan microbial catabolites, including indole, which in turn results in increased susceptibility of mice to KP infection.

Administration of mice with indole or probiotic cocktail reversed alcohol effects on KP infection rate and also mitigated some inflammatory parameters. The involvement of aryl hydrocarbon receptor AhR in the process is anticipated, because addition of AhR antagonist CH223191 abrogated the attenuation of infection and inflammation by indole. Overall, the study is rigorously elaborated; however, there is a number of specific a conceptual issues that significantly dampen the enthusiasm for the paper.

1/ There is no mention in abstract and introduction, on the roles on tryptophan intestinal microbial catabolites in physiology and patho-physiology, but the paper is focused on these compounds.

Response: We have added discussion regarding the role of microbial catabolites in physiology and patho-physiology to the introduction and abstract.

2/ The authors say (in abstract), that alcohol impairs the production of microbial catabolites, "specifically indole". However, in Fig 5, they show that alcohol also drastically diminishes the levels of I3-sulphate, I3-acetate and I3-lactate.

Response: We have amended this to state "tryptophan catabolites".

3/ There are concerns about intestinal/fecal concentration of indole, reported in the study. P12 (discussion) the authors cite paper, where are reported indole fecal concentration in humans about 250-1100 mM. Cecal concentrations of indole (Fig 5) were determined to be around 10 nmol/g (i.e. approx. 10 uM). Mice were administered with 20 mg/kg/dose of indole. It should be clarified how are related these three concentrations, and also using common units to allow comparison is recommended.

Response: Indole has been administered to mice at the following concentrations;

1. 3 μ mol/20 g
 - a. doi:[10.1096/fj.201800544](https://doi.org/10.1096/fj.201800544)
2. 20 mg/kg
 - a. <https://doi.org/10.1080/19490976.2016.1156827>
3. 50 mg/kg
 - a. <https://doi.org/10.1002/hep.31115>
4. We choose to give mice 200 uL of 17 mM Indole (or 20 mg/kg).

Cecal indole levels in mice have been reported to range between 200 and 400 μM , while indole-3-acetate were less abundant, with concentrations ranging from 10–40 μM (<https://doi.org/10.1124/mol.113.091165>, however other studies have detected intestinal indole-3-acetate and indole-3-sulfate at 10 pmol/mg and 250 pmol/mg, respectively (<http://dx.doi.org/10.1136/gutjnl-2018-317232>). Our measured values are in-between these ranges. However, cecal or fecal indole levels are most likely highly dependent on diet, mouse strain, and disease condition. Regardless, we have amended the methods section and the text throughout to include a molar concentration in each spot for ease of comparison.

4/ Indole was described as human-specific ligand of AhR (Hubbard et al 2015 Sci Rep). An affinity of indole against mouse AhR was recently reported, but IC₅₀ in competitive ligand-binding assay was higher than 1 mili M. Also EC₅₀ in reporter assays in human cells by indole was about 1.5 mili M (Vyhlidalova et al 2020 Int J Mol Sci). These concentrations, both in human and mice models, are apparently much higher than those used in the study by Samuelson et al. How to explain? Fig 5A – indole cecal levels are restored in indole-treated mice back to approx. 10 micro M, hence too low to activate AhR.

Response: We have performed additional *in vivo* and *in vitro* assays to assess AhR activation by indole in mice. First in cell culture experiments using mouse AhR reporter cells (doi:10.1016/j.chemosphere.2008.08.015) we found that indole activates AhR at 5-20 μM concentrations. In addition, we added CH-223191 at a concentration of 30 μM to the indole treated wells and found indole-mediated luciferase activity was blocked by the addition of the AHR antagonist, see New Figure 7. In addition to assessing the effects of indole alone, we also isolated microbial products from mice in each of the treatment conditions and assessed their ability to activate AhR using the *in vitro* reporter assay. We found that all of the microbial products stimulated AhR above untreated cells, but microbial products isolated from alcohol-fed mice or mice given the AHR inhibitor exhibited significantly reduced levels of AhR activation compared to microbial products isolated from pair-fed or indole treated mice, see New Figure 7.

Finally, we also assessed AhR activation *in vivo* via qPCR. Specifically, we assessed Ahr, Cyp1a1, and Il-22 mRNA levels in the lungs and the intestinal track of each treatment group and found the Ahr expression, as well as Cyp1a1 and Il-22 were significantly reduce in alcohol-fed animals compared to control animals, see New Figure 6.

5/ The involvement of AhR in the effects observed by indole is not convincingly evidenced in the paper: (i) There is absent any proof of AhR activity (basal, induced, inhibited), e.g. by AhR target gene expression etc.; (ii) The use of AhR antagonist CH223191 is inappropriate and insufficient approach to demonstrate AhR involvement. This antagonist is ligand selective, e.g. it antagonize effects by TCDD but not by bezo-a-pyrene. Hence, dose-response effects of indole (itself and/or combined with CH223191), either in cell-based assay, or *in vivo* gene expression should be shown.

Response: Please see comment for point 4 above.

6/ Conceptual: There is presented that alcohol decreases indole levels in cecum, and indole supplementation restores indole levels. It is kind of tautology, when showing that a drop in concentration of the compound is reversed by addition of the same compound. There should be demonstrated clear link between plasma and intestinal/fecal levels of indole in control and alcohol-fed mice and indole-fed mice. It is essential to distinguish between mechanistic base of indole effects, or simple external replenishment of depleted indole levels.

Response: We were unable to measure indole levels in the serum; however we have added this point to the discussion, see revised discussion section.

Reviewer #2 (Remarks to the Author):

1. Fig 1b, 1c, 3b, 3c: There are a number of data points along the x axis. Did these animals have undetectable *K. pneumoniae* burden? Does the extent of colonization of the probiotic relate to the amount of *K. pneumoniae* burden, i.e., those animals with a higher lung burden had a corresponding lack of colonization by the administered probiotics?

Response: Yes, the data on the x-axis have undetectable *K. pneumoniae* burden. We do not believe that the burden of *K. pneumoniae* is related to probiotic colonization. First, the heat-killed probiotic also provides a similar level of protection as the live probiotics. Suggesting that probiotics do not need to colonize the GI tract to assert beneficial effects. Strain-level qPCR for each individual *Lactobacillus* could be ran to better assess the correlation between probiotic and burden of *K. pneumoniae*, however we feel that this does not significantly add to the current data set.

2. The authors should quantify intestinal and pulmonary IL-22 production to confirm active AhR signaling.

Response: Please see comment for point 4 from review 1 above.

3. In figure 6c, I do not see any *Lactobacilli* in the probiotic-treated groups. The probiotic mixture contains 2-3 *Lactobacilli*, so can the authors comment on why not detected? Can they use 16S qPCR to quantify whether these species engrafted at all (also relevant for point #1).

Response: Actually, all of the groups have some detectable level of *Lactobacillus* at a genus level they are just not differentially abundant between the groups, as we group all sequences belonging to that genus into a single unit. However, we further evaluated the 16s sequencing data at a species level and found that *Bifidobacterium animalis* and *Lactobacillus reuteri* (two species present in the probiotic cocktail) were only detected in probiotic treated and the heat-killed probiotic groups, please see the new Supplemental

Figure 4a. In addition, both of these species were significantly different based on species level differential abundance analysis, see new figure 8c.

4. For figures 6d-f, it is surprising that no inferred pathways were significantly changed with live probiotics, whereas this change is observed with heat-killed probiotics. Can the authors comment? Moreover, it appears that CH-223191 (i.e. AhR inhibition) does not alter any of these inferred bacterial pathways either, which presumably would be impacted by altered host AhR signaling.

Response: These are interesting points. We have added some additional text to the discussion section to address/speculate on these points, see revised discussion section.

5. Heat killed probiotics also have protective effects, which suggests (also mentioned by authors) an additional mechanism of protection. Can the authors comment on what it could be? Could it also be dependent on AhR signaling? While not necessary for this manuscript, the authors should consider a future study in which heat-killed probiotics administered arm is also treated with CH223191.

Response: We have added some additional text to the discussion section to address/speculate on these points, please see revised discussion section. In addition, in the course of performing additional experiments to address other comments we added the heat-killed probiotics treated with CH223191 group, please see the results in New Figure 5, 6 and 7.

6. These studies indicate that prophylactic probiotics and indole administration (and AhR activation) can protect from binge-drinking associated *K. pneumoniae* pathogenesis. However, can these be used as therapeutic interventions? Here, comments + citations are sufficient.

Response: These are interesting points. We have added some additional text to the discussion section to address/speculate on these points, see revised discussion section.

7. Minor:

- v In the Abstract line 34, I think the authors mean “mitigate” and not “migrate”.
- v Please describe how bacteria were heat killed.
- v Make data points in the PcoA plot in 6b larger as they’re hard to see.
- v Please check figure legends for lower case p in pneumoniae.

Response: We have corrected all minor points.

Reviewer #3 (Remarks to the Author):

This study investigates the impact of indole and probiotic treatment on *K. pneumoniae* lung infection in alcohol-fed mice. Previously, the authors had shown that alcohol consumption associated dysbiosis increases the susceptibility of mice to *K. pneumoniae* infection in the lungs. Overall, the results shown here support the premise that raising the levels of microbiota metabolites, specifically tryptophan catabolites in the intestine, could improve immune cell counts in the lung while reducing *K. pneumoniae* burden. Using a chemical inhibitor, the authors conclude that the effects of indole are mediated by the AhR. This conclusion requires additional evidence, as the authors do not provide direct evidence of AhR activation by indole in target tissues. Further, several aspects of the study require further explanation. The discussion could also be improved by tying together the observations regarding indole and probiotic treatment. Some conclusions, e.g., cytokine modulation, appear to be speculative, as the paper does not present these data. Specific comments follow.

1. Please comment on the distribution of responses between animals. It seems that some animals in the treatment groups (indole, probiotic) had zero *K. pneumoniae* burden in both the lung and spleen, and this is what decreased the group averages. In treated animals that showed *K. pneumoniae* burden, the CFUs seem comparable to the alcohol-fed controls.

Response: Yes, some of the mice have undetectable *K. pneumoniae* burden. However, every mouse was given the same exact inoculation, suggesting that indole, probiotics are improving host defense. While, the burden in some of the treated mice is similar to alcohol-fed mice the response to infection is still improved, see immunological data and ELISA data. In addition, in our recent experiments we found that indole and probiotics still reduced *K. pneumoniae* burden, even when all the mice have detectable CFU.

2. In addition to IFABP, have the authors measured LPS in circulation? Translocation of bacterial endotoxin would more directly indicate a weakened barrier function in the context of an inflammatory response. Some studies, e.g., Lau et al., (2016) have suggested that IFABP is not a reliable marker of intestinal barrier dysfunction. Further, IFBAP is expressed in epithelial cells of the small intestine, whereas the microbiota analyses in this paper focused on the cecum.

Response: We have found IFABP to be reliable for barrier dysfunction in our model system, as it mirrors results from FITC-dextran challenge experiments. In addition, LPS is challenging to measure in our model system as we also see disseminated infection with *Klebsiella* (a Gram- organism). As such, it is impossible to distinguish between intestinal derived LPS and LPS derived from the infection, as all of the samples are collected post infection.

3. I have several concerns regarding the AhR experiments. What concentration of indole in the various tissues, i.e., lung, intestine, spleen, etc.? What is the evidence that indole activated the AhR as a ligand? Have the authors measured a panel of AhR genes, e.g., CYPs, in the relevant tissues? The text implies that IL-22 production should

be modulated. Have the authors measured the effect of indole on IL-22 production in intestinal immune cells (e.g., Th17 cells)? Along this vein, have the authors checked that CH-223191 is detected at sufficiently high (inhibitory) concentrations in the target tissues and that expression of AhR genes is down-regulated?

Response: While we are unable to reliably detect indole or CH-223191 in tissues. We did perform additional experiments to address the role of AhR in our model system. Please see comment for point 4 from review 1 above and the data presented in New Figure 5, 6 and 7.

4. The connection between CH-223191 and probiotic treatment is unclear. The authors point out that live and heat-killed probiotics elicited similar responses, suggesting that metabolic activity of the probiotics is not necessary. Does this mean that the heat-killed probiotics contained AhR ligands?

Response: That is one possibility, or perhaps the heat-killed bacteria improved or enhanced the microbiota's ability to produce AhR ligands, by supporting beneficial or suppressing detrimental microorganism. We have added some additional text to the discussion section to address/speculate on these points, please see revised discussion section.

5. Results shown in figure 5 indicate that alcohol feeding depletes several tryptophan catabolites in the cecum, which are restored if the treated animals receive indole or a probiotic cocktail. Given the above question, it would be interesting to know if the heat-killed probiotic also raised the levels of the metabolites. What are the levels of these metabolites in the lung and spleen?

Response: While we are unable to reliably detect indole or CH-223191 in tissues, we have added some additional text to the discussion section to address/speculate on these points, please see revised discussion section.

6. I found the description of the microbiome sequencing results difficult to follow. Figure 6A includes a symbol (plus) that does not map to any experimental group. These plots show N = 5, rather than N = 10. It would be interesting to know if the treated animals in this figure correspond to the animals that showed minimal (~zero). The probiotic group in Figure 6A projects closer to the alcohol-fed group than to the pair-fed group. However, the text states that the beta diversity is significantly different between these two groups. Curiously, the heat-killed probiotic group projects closer to the indole and pair-fed group than the live probiotic group.

Response: We have retooled the microbiome section.

7. The PCA scatter plot shown in Figure 6B is very difficult to read. The symbols are small and have similar colors. What is meant by "inferred functional capacity?"

Response: We have improved Figure 6B. Inferred functional capacity refers to predict microbial gene pathways that are different between groups. This is done using a bioinformatic approach based on the results of 16s sequencing. It is inferred because 16s sequencing data searched against known whole genome sequences and the abundance of the specific microbial gene pathways is inferred based on the known information from each genome.

8. Figure 6C indicates that the relative abundance of the genus *Klebsiella* is substantially higher in several treatment groups compared to the pairwise-fed control, including indole, probiotic, and heat killed probiotic. Does this genus abundance count exclude the pathogen? Further, why would the *Bifidobacterium* count decrease in probiotic treated animals when the cocktail includes this genus? The Results section makes no mention of these observations. Also, given the large number of changes shown in the figure, the following statement is confusing: “In most instances treatment with indole, probiotics, or heat-killed probiotics restored the relative abundance of OTUs to levels similar to those seen in pair-fed control mice ...”

Response: We apologize for the confusion; we have reworked this entire section to make it easier to read.

9. Have the authors measured representative metabolites from the pathways shown in Figures 6D-F, e.g., tyrosine, phenylalanine, tryptophan, valine, etc.? Having the metabolite data could corroborate the PICRUST2 results.

Response: We have not performed the metabolomic study to confirm the results from picrust2, we feel that this is beyond the scope of the current paper.

10. If neither microbiome measure (alpha or beta diversity) is a significant predictor of *K. pneumoniae* burden, then what is the link in the gut-lung axis the authors describe?

Response: We have previously published that adoptive transfer of the microbiota from an alcohol-fed mouse into an alcohol-naïve mouse increases susceptibility to *K. pneumoniae*. In addition, just because alpha and beta diversity do not correlate with burden (in a small subset of the all the mice tested) other microbiome metrics may in fact correlate with burden. For example, it is highly likely that specific taxa, metabolites or groups of taxa/metabolites correlate with *K. pneumoniae* burden. However, these analyzes are beyond the scope of this paper and would require additional animal models to validate the identified taxa.

11. I would have liked to see a discussion of the different effects of probiotic and indole treatments. The AhR inhibitor experiments suggest that CH-223191 attenuated some of the effects of probiotic treatment, e.g., SPD-1, immune cell counts (Figure 4). This suggests that there could be a shared mechanism, but the two treatments are discussed separately.

Response: we have added some additional text to the discussion section to address/speculate on these points, please see revised discussion section.

12. Additional controls are needed to establish that indole is acting through the AhR (see point #3). The possibility that indole treatment restores cytokine/chemokine levels is intriguing, but this seems to be speculative, as these data are not part of this study. Similarly, the authors did not directly assess if indole treatment impacted immune cell programming.

Response: We did perform additional experiments to address the role of AhR in our model system. Please see comment for point 4 from review 1 above and the data presented in New Figure 5, 6 and 7.

13. The timing of treatment is interesting. What would happen if the animals were exposed to indole and/or probiotic post-infection rather than pre-infection (which would more closely mimic the administration of a therapeutic)?

Response: These are interesting points. We have added some additional text to the discussion section to address/speculate on these points, please see revised discussion section.

Reviewers' comments:

Reviewer #1 (Remarks to the Author):

The authors addressed my comments only partly. In particular, a part describing the activation of the AhR by indole (Figure 7) needs improvement. Application of 20 micro M indole causes increase of luciferase activity approx. 2-fold, which is virtually at the baseline level. As mentioned and referenced in my initial review, IC50 values in ligand binding assays, or EC50 values in reporter assays for indole are in millimolar range; then reaching activation approx 1000-fold or so. In addition, Figure 7a does not show any reference AhR agonist, e.g. TCDD, to evidence dynamic range of the assay. Other issues including the inappropriateness of CH223191 use or conceptual issues were inadequately addressed.

Reviewer #2 (Remarks to the Author):

I appreciate the authors addressing my concerns. In particular, experiments using AhR inhibitor in heat-killed probiotic supplemented animals, in Fig 5 and 6 which demonstrate that intact AhR signaling is essential for immune cell recruitment in alcohol-related Kp pneumonia.

I recommend accepting this paper following the following points being addressed:

- Fig 8b: are any of these mice inoculated with Kp, or are these indigenous Klebsiella? I realize this is genus-level resolution, but was still surprised that probiotic treated (both regular and heat-killed) have increased intestinal Klebsiella OTUs following alcohol consumption, compared to other groups. This kind of is opposite of what is seen in the lung.
- Similarly, it appears that alcohol binge-drinking reduces the levels of *L. reuterii* as seen in fig 8c, which seems at odds with Supp fig 5a. My read of the text indicates this is the same data analyzed with DESeq2. Please comment.
- Please elaborate Supp Fig 4A.
- Supp 5D is mislabeled

Reviewer 1: The authors addressed my comments only partly. In particular, a part describing the activation of the AhR by indole (Figure 7) needs improvement. Application of 20 micro M indole causes increase of luciferase activity approx. 2-fold, which is virtually at the baseline level. As mentioned, and referenced in my initial review, IC50 values in ligand binding assays, or EC50 values in reporter assays for indole are in millimolar range; then reaching activation approx 1000-fold or so. In addition, Figure 7a does not show any reference AhR agonist, e.g. TCDD, to evidence dynamic range of the assay. Other issues including the inappropriateness of CH223191 use or conceptual issues were inadequately addressed.

Response:

We have performed additional cell reporter experiments to examine a more dynamic range of concentrations. In addition, we included the reference AhR agonist, TCDD. Similar to our previous results we show that indole activates AhR and that CH223191 blocks indole-mediated activation (New Figure 8). Similarly, TCDD activates AhR in a dose dependent manner and is blocked by CH223191 (New Supplemental Figure 4). Our results are similar to other published data that shows mouse AhR is activated by indole, but that TCDD is a much more potent activator of murine AhR. In fact, indole activates AhR at ~20% of 10^{-10} M TCDD. We have added additional information regarding these findings to the discussion (see lines 364-373). Specifically, we hypothesize that TCDD causes a sustained and inappropriate activation of AhR which may lead to detrimental effects. In contrast, indole induces transient lower (than TCDD) activation of AhR, which we believe to be critical for maintaining normal cellular homeostasis. Additional research is needed to understand the balance between beneficial and detrimental effects of AhR signaling. Further, more information is needed to understand if the positive and negative effects of AhR signaling are either ligand or dose-dependent. Further, our *in vivo* data supports a clear biologically relevant effect of indole and AhR (regardless of an only approx. 2-fold increase), as we show that in both the lungs and gi tract of mice treated with indole that the genes down stream of AhR signaling (i.e., Cyp1a1, and IL-22) are significantly increased compared to vehicle treated mice and that these responses are impaired by CH223191 treatment.

Finally, in the references provided by the reviewers, the authors showed that the EC50 for indole in a human cell line, and the IC50 in a mouse line was indeed in the millimolar range. However, these data do not refute the ability of indole to bind to and activate AhR at micromolar doses. While it is true that indole must be in the millimolar range to outcompete TCDD, a potent ligand, for AhR binding this does not mean that indole cannot bind to and activate AhR at lower doses in murine systems, which is supported by our data.

We respectfully disagree with the reviewer's comments regarding the use of CH223191. This AhR inhibitor has been used in numerous *in vitro* and *in vivo* studies to block ligand mediated activation of AhR, including indole-mediated activation. In addition, it has been previously reported that indole-3-aldehyde (an indole derivative) at micromolar

concentrations activates AhR, with both CH223191 and AhR siRNA used for confirmation (see a select group of refs below). In fact, based on the reviewer's primary suggestions we showed in both *in vitro* and *in vivo* studies that CH223191 blocks indole-mediated activation of AhR using reporter cells, as well as qPCR analysis of genes down stream of AhR (Cyp1a1, and IL-22).

1. <https://doi.org/10.1002/hep.31115>
2. <https://doi.org/10.1124/mol.105.021832>
3. <https://doi.org/10.1016/j.jdermsci.2020.10.004>

We also reanalyzed indole concentrations using a more sophisticated method to determine cecal, serum, and lung tissue indole concentrations (New Figure 7), which support our current findings, and help address some of the reviewer's conceptual concerns. Specifically, indole levels are highest in the indole treated mice, even in the presence of CH223191. Using this new method indole levels were found to be ~45 nmol/g (45 uM) in the cecum, ~0.189 umol/mL (189 uM) in the serum, and ~0.133 umol/mL (133 uM) in the lung tissue. All of these levels of indole cause a significant increase in AhR activation in our cell reporter assay.

Reviewer 2: I appreciate the authors addressing my concerns. In particular, experiments using AhR inhibitor in heat-killed probiotic supplemented animals, in Fig 5 and 6 which demonstrate that intact AhR signaling is essential for immune cell recruitment in alcohol-related Kp pneumonia.

I recommend accepting this paper following the following points being addressed:

-- Fig 8b: are any of these mice inoculated with Kp, or are these indigenous Klebsiella? I realize this is genus-level resolution, but was still surprised that probiotic treated (both regular and heat-killed) have increased intestinal Klebsiella OTUs following alcohol consumption, compared to other groups. This kind of is opposite of what is seen in the lung.

Response: Mice were inoculated with *K. pneumoniae* (strain 43816, serotype 2; American Type Culture Collection, Manassas, VA), which is not the indigenous Klebsiella. We do not have the resolution to determine if the intestinal Klebsiella strain is the same as the Klebsiella used for infection, but we hypothesize that it is not the same strain. Klebsiella is a normal inhabitant of the intestinal track in mice, as such it is likely that different treatments will affect the growth of native Klebsiella. We have added some additional text to the discussion to elaborate on this finding.

-- Similarly, it appears that alcohol binge-drinking reduces the levels of *L. reuterii* as seen in fig 8c. which seems at odds with Supp fig 5a. My read of the text indicates this is the same data analyzed with DESeq2. Please comment.

-- Please elaborate Supp Fig 4A.

-- Supp 5D is mislabeled

Response: Figure 8c and Supplemental 5a show the same data, however in Figure 8c the data is graphed as the differential abundance between the treatment groups and vehicle treated alcohol-fed mice. That is to say that in Figure 8 c it shows that alcohol-fed mice have reduced the levels of *L. reuterii* compared to probiotic or heat-killed probiotic treated mice. While Supplemental Figure 5a shows the absolute abundance of *L. reuterii* in every group, or in other words *L. reuterii* is only detected in the probiotic or heat-killed probiotic treated mice.

We have provided additional information on Supplemental 4a (now 5a). Supplemental Figure 5d (now 6d) is labeled correctly, however the colors are different from Supplemental Figure 4 (now 5), as such we have replaced supplemental figure 5D to the same colors scheme.

Reviewer #3 Comments:

In this revision, the authors have thoughtfully addressed a majority of this reviewer's concerns. In particular, the authors provide new data (figure 6) showing that AhR, Cyp1a1 and IL-22 are all upregulated in the lung upon indole treatment. The immune cell profiles shown in figure 5 are also noteworthy. The authors also show that cecal contents from alcohol fed mice show reduced activation of AhR. A few concerns remain, however, regarding the indole results and their connection to the probiotic treatment outcomes. The microbiome results also need additional clarification.

1. I am still unclear as to why indole or the inhibitor cannot be detected reliably in the target tissues when the authors have shown that they can measure indole in the cecum. The data shown in supplemental figure 3 suggests that the assay the authors used can detect a few uM concentrations of indole. This would be below the level needed for AhR activation. The results of figure 6 suggest that indole should be present at a higher concentration the lungs.

Response: We reanalyzed indole concentrations using a more sophisticated method to determine cecal, serum, and lung tissue indole concentrations (New Figure 7), which support our current findings, and help address some of the reviewer's conceptual concerns. Specifically, indole levels are highest in the indole treated mice, even in the presence of CH223191. Using this new method indole levels were found to be ~45 nmol/g (45 uM) in the cecum, ~0.189 umol/mL (189 uM) in the serum, and ~0.133 umol/mL (133 uM) in the lung tissue. All of these levels of indole cause a significant increase in AhR activation in our cell reporter assay. Finally, we are not aware of any established methods to evaluate CH223191 levels in tissues.

2. Figure 6c shows that the heat-killed probiotic is dramatically more potent in upregulating lung IL-22 expression compared to the live probiotic. Surprisingly, the live probiotic does not show any effect. However, figure S3 shows that the cecal indole levels are nearly identical for mice given indole or probiotic. I agree with the authors' point that the probiotic (live or heat-killed) likely acts not only through the AhR but additional pathways; but, in the case of IL-22 expression, the antagonist completely

blunted the effect of the probiotic. These are confusing results that warrants further explanation. For example, would it possible that the heat-killed probiotic stimulated the production of a different AhR ligand instead of indole?

Response: We have added more information to the discussion regarding this question (See lines 354-361). We, like the reviewer, believe that the heat killed probiotic is increased other AhR ligands other than indole to a much greater extent than the live probiotics alone. Furthermore, ROR γ t expression alone may induce transcription of the IL22 gene, however ROR γ t and AhR often act synergistically to promote IL-22 expression. As such, it is likely that heat-killed probiotics have a much greater effect on ROR γ t.

3. The authors state that alcohol and probiotic treated mice have significantly different beta diversity, but the plots of Bray-Curtis dissimilarity in figure 8 show overlap between these two conditions. Unless figure 8 was not used to compare beta diversity between groups?

Response: Using the plot of the data to make comparisons to the statistics is hard. For example, the plot only represents the data in 2 dimensions, however the data is multi-dimensional, so if you rotate the data along one of the axes, you will see additional separation between probiotic and alcohol-fed animals. All stats were generated from the complete data sets, and p values were calculated for pair-wise comparisons between alcohol-fed animals and the indicated group. All p values obtained were then corrected for multiple comparisons using FDR.

4. In mice, the genus Akkermansia is thought to be dominated by a single species, *A. muciniphila*. Yet figure 6, panels b and c show different results between genus and species level fold-changes. Have the authors detected another Akkermansia species?

Response: No, we do not believe that is the case. The difference between panel b and c is that panel C only uses sequences that can be readily identified down to the species level (i.e., data only includes sequences identified as *Akkermansia muciniphila*), while panel B uses all the sequences that can be identified at the genus level (i.e., data includes sequences identified as *Akkermansia* spp, or *Akkermansia muciniphila*). For every sequencing run you will get multiple OTUs that map to a genus, and only some of those OTUs can be identified at the species level.

5. Regarding the authors response to this reviewer's point #10, I agree that specific taxa and/or metabolites could correlate with *K. pneumoniae* burden, even if there is no correlation with alpha or beta diversity. The authors report on several significantly altered taxa in treated mice that show reduced *K. pneumoniae* burden – were any of these correlated with bacterial counts in the lung?

Response: We have performed additional analysis to evaluate the correlation of the differentially abundant taxa and pulmonary bacterial burden. 7 out of the 27 differentially

abundant bacterial genera were significantly associated with Klebsiella burden. We have added this information to the results section, see lines 258 to 263.

Reviewers' comments:

Reviewer #1 (Remarks to the Author):

In this revised manuscript, the authors addressed properly my comments. Therefore, I recommend this publication for acceptance in Communications Biology.

Reviewer #2 (Remarks to the Author):

The authors have sufficiently addressed my concerns. I recommend the editor accept this paper.

Reviewer #3 (Remarks to the Author):

The authors have addressed most of this reviewer's concerns. I do have one remaining question, however, which the authors only partially answer, even in this second revision.

The authors report on a more sophisticated indole assay, and a new figure shows indole concentrations in tissues as low as a few ng/ml. Still, indole is not detected in lung tissues of probiotic (live or heat inactivated) animals. Moreover, probiotic treatment did not substantially elevate serum indole levels. An alternative, microbiota derived AhR agonist was not measured. The authors acknowledge that often, there is no statistical difference between probiotic treated mice and probiotic treated mice given CH-223191, whereas the inhibitor always had an effect in indole treated mice. I wonder then: what is the link between indole, AhR and the probiotic treatments? Taken together, these observations do not support an AhR mediated mechanism for the probiotic's effects in the lungs. Consequently, the claim that there is a shared mechanistic pathway (as stated in the discussion) is also not supported. My critique does not take away from the robust phenotypes reported in this manuscript. However, based on the data presented, I think it is premature to conclude an AhR dependent mechanism for the probiotics.

Reviewer #1 (Remarks to the Author):

In this revised manuscript, the authors addressed properly my comments. Therefore, I recommend this publication for acceptance in Communications Biology.

Reviewer #2 (Remarks to the Author):

The authors have sufficiently addressed my concerns. I recommend the editor accept this paper.

Reviewer #3 (Remarks to the Author):

The authors have addressed most of this reviewer's concerns. I do have one remaining question, however, which the authors only partially answer, even in this second revision.

The authors report on a more sophisticated indole assay, and a new figure shows indole concentrations in tissues as low as a few ng/ml. Still, indole is not detected in lung tissues of probiotic (live or heat inactivated) animals. Moreover, probiotic treatment did not substantially elevate serum indole levels. An alternative, microbiota derived AhR agonist was not measured. The authors acknowledge that often, there is no statistical difference between probiotic treated mice and probiotic treated mice given CH-223191, whereas the inhibitor always had an effect in indole treated mice. I wonder then: what is the link between indole, AhR and the probiotic treatments? Taken together, these observations do not support an AhR mediated mechanism for the probiotic's effects in the lungs. Consequently, the claim that there is a shared mechanistic pathway (as stated in the discussion) is also not supported. My critique does not take away from the robust phenotypes reported in this manuscript. However, based on the data presented, I think it is premature to conclude an AhR dependent mechanism for the probiotics.

Response:

We agree that the effects/mechanisms by which probiotics provide protection are likely not solely dependent on AhR signaling, and most likely influence multiple pathways. We have amended the discussion regarding the AhR dependent mechanism for probiotics and added some additional details regarding the potential connections between probiotics, indoles, and AhR signaling, see below (lines 410-438). In addition, we have modified the title of the paper to better represent the outcomes of the paper.

"Treatment of alcohol-fed mice with the probiotic cocktail, heat-killed probiotics, or with indole lead to a significant reduction in pulmonary bacterial burden. Suggesting that alcohol-mediated changes to the intestinal microbiota are critical mechanistic pathway by which alcohol impairs pulmonary host defense against bacterial pneumonia. However, there were specific effects on immune cell profiles, the intestinal microbial

communities, as well as the involvement of AhR signaling unique to each treatment condition. For example, AhR inhibition leads to mitigation of every beneficial effect following indole treatment, suggesting that indole mainly acts as an AhR agonist. However, AhR inhibition does not mitigate all the beneficial effects of probiotics or heat-killed probiotics treatments. Specifically, AhR inhibition does not impair the trafficking of immune cells to the lung in response to pneumonia in animals that received probiotic supplementation. While AhR inhibition does blunt the effects of probiotics on bacterial burden, immune cell numbers, epithelial permeability, and gene expression there is often no statistical difference between probiotic treated mice and probiotic treated mice given CH-223191, while AhR inhibition in indole treated mice is always significantly different from indole treated alone. This strongly suggests that probiotics or heat-killed probiotics act via multiple pathways. However, the lack of an effects of probiotics on AhR signaling may be due to several reasons. First, microbial tryptophan catabolites have been shown to have both agonistic and antagonistic effects against other microbial and environmental AhR ligands.⁵⁴ For example, indole-3-acetamide and indole-3-pyruvate exhibit dual effects on ligand-activated AhR. Each of the indole metabolites antagonize TCDD but potentiate the agonistic effects of BaP and FICZ.⁵⁴ Given this data, it is possible that probiotic treatments results in a shift in the production and/or amount of different indole metabolites such that one metabolite acts as an AhR antagonistic in our model system. This may help to explain the intermediate effects of CH-223191 treatment that we observed in the probiotic or heat-killed treated mice. Finally, the kinetics of indole production and conversion may be different between probiotic treated mice and indole treated mice, as we only evaluated mice at one time point post infection. As such, it is possible that the probiotic-mediated effects on AhR signaling occur early then we examined. These data strongly support the need for a more complete understanding of the link between probiotics, indole, and AhR signaling in the context of host-defense against alcohol-associated pneumonia.”